EMBO
Molecular Medicine

# Coenzyme Q deficiency causes impairment of the sulfide oxidation pathway

Marcello Ziosi[1,†], Ivano Di Meo[2,†], Giulio Kleiner[1], Xing-Huang Gao[3], Emanuele Barca[1,4], Maria J Sanchez-Quintero[1], Saba Tadesse[1], Hongfeng Jiang[5], Changhong Qiao[5], Richard J Rodenburg[6], Emmanuel Scalais[7], Markus Schuelke[8], Belinda Willard[9], Maria Hatzoglou[3], Valeria Tiranti[2,*,‡] & Catarina M Quinzii[1,‡,**]

## Abstract

Coenzyme Q (CoQ) is an electron acceptor for sulfide-quinone reductase (SQR), the first enzyme of the hydrogen sulfide oxidation pathway. Here, we show that lack of CoQ in human skin fibroblasts causes impairment of hydrogen sulfide oxidation, proportional to the residual levels of CoQ. Biochemical and molecular abnormalities are rescued by CoQ supplementation *in vitro* and recapitulated by pharmacological inhibition of CoQ biosynthesis in skin fibroblasts and ADCK3 depletion in HeLa cells. Kidneys of *Pdss2*[kd/kd] mice, which only have ~15% residual CoQ concentrations and are clinically affected, showed (i) reduced protein levels of SQR and downstream enzymes, (ii) accumulation of hydrogen sulfides, and (iii) glutathione depletion. These abnormalities were not present in brain, which maintains ~30% residual CoQ and is clinically unaffected. In *Pdss2*[kd/kd] mice, we also observed low levels of plasma and urine thiosulfate and increased blood C4-C6 acylcarnitines. We propose that impairment of the sulfide oxidation pathway induced by decreased levels of CoQ causes accumulation of sulfides and consequent inhibition of short-chain acyl-CoA dehydrogenase and glutathione depletion, which contributes to increased oxidative stress and kidney failure.

**Keywords** coenzyme Q; CoQ10; Pdss2; SQR; sulfides
**Subject Categories** Genetics, Gene Therapy & Genetic Disease; Metabolism

See also: **M Luna-Sánchez et al**

## Introduction

Coenzyme Q (CoQ), or ubiquinone, is a lipid antioxidant and electron carrier, present in all cell membranes, and involved in many biological processes (Bentinger *et al*, 2010). Mutations in genes encoding proteins involved in CoQ biosynthesis cause primary CoQ deficiency, an autosomal recessive mitochondrial disorder associated with five major clinical phenotypes: (i) encephalomyopathy, (ii) severe infantile multi-systemic disease, (iii) cerebellar ataxia, (iv) isolated myopathy, and (v) steroid-resistant nephrotic syndrome (Desbats *et al*, 2015). This heterogeneity in the clinical presentation suggests that multiple pathomechanisms may exist. Because of its essential function as electron carrier from mitochondrial complexes I and II to complex III as well as an endogenous antioxidant, the effects of CoQ deficiency on mitochondrial bioenergetics and oxidative stress have been assessed *in vitro* (Lopez *et al*, 2014) and *in vivo* (Licitra & Puccio, 2014). Increased oxidative stress and ATP depletion have been described as initial events promoting mitochondrial-mediated apoptotic cell death (Lopez *et al*, 2014) and/or autophagy (Rodriguez-Hernandez *et al*, 2009).

CoQ-deficient fission yeast have been shown to produce high sulfide levels (Zhang *et al*, 2008), and more recently, proteomics studies revealed decreased protein levels of SQR in CoQ-deficient mice carrying a missense mutation in COQ9, a protein involved in the biosynthesis of CoQ through regulation of the levels of the hydroxylase COQ7 (Lohman *et al*, 2014; Luna-Sánchez *et al*, 2015). However, whether the $H_2S$ metabolism is impaired in CoQ deficiency, and whether $H_2S$ metabolism impairment has a role in the pathogenesis or the progression of this disorder, has not yet been investigated.

1  Department of Neurology, Columbia University Medical Center, New York, NY, USA
2  Unit of Molecular Neurogenetics, IRCCS Foundation Neurological Institute "Carlo Besta", Milan, Italy
3  Department of Genetics and Genome Sciences, Case Western Reserve University, Cleveland, OH, USA
4  Department of Clinical and Experimental Medicine, University of Messina, Messina, Italy
5  Irving Institute for Clinical and Translational Research, Columbia University Medical Center, New York, NY, USA
6  Department of Pediatrics, Radboud Center for Mitochondrial Medicine (RCMM), RadboudUMC, Nijmegen, The Netherlands
7  Division of Paediatric Neurology, Department of Paediatrics, Centre Hospitalier de Luxembourg, Luxembourg City, Luxembourg
8  Department of Neuropediatrics and NeuroCure Clinical Research Center, Charité-Universitätsmedizin Berlin, Berlin, Germany
9  Mass Spectrometry Laboratory for Protein Sequencing, Learner Research Institute, Cleveland Clinic, Cleveland, OH, USA
  *Corresponding author. Tel: +39 02 23942633; Fax: +39 02 23942619; E-mail: valeria.tiranti@istituto-besta.it
  **Corresponding author. Tel: +1 212 305 1637; Fax: +1 212 305 3986; E-mail: cmq2101@cumc.columbia.edu
  †These authors contributed equally to this work
  ‡These authors contributed equally to thus work

    

Sulfide ($H_2S$) is a gas modulator (like nitric oxide and carbon monoxide) produced endogenously via the cysteine catabolism by the cytoplasmic enzymes cystathionine-β-synthase (CBS), cystathionine-γ-lyase (CSE, CTH), and 3-mercaptopyruvate by 3-mercaptopyruvate sulfurtransferase (3-MST). CBS is predominantly active in the nervous system, kidney, and liver; CSE in liver, intestine, smooth muscle, and pancreas (Stipanuk & Beck, 1982; Hosoki *et al*, 1997; Zhao *et al*, 2001; Enokido *et al*, 2005; Tiranti & Zeviani, 2013; Gao *et al*, 2015), and 3-MST, a mitochondrial and cytoplasmic enzyme, is active mainly in the nervous system (in particular cerebellum), kidney, and vascular endothelium, where it exerts an important cytoprotective role (Steegborn *et al*, 1999; Shibuya *et al*, 2009, 2013).

$H_2S$ physiologically modifies cysteine residues in numerous proteins by S-sulfhydration, affecting their functions (Mustafa *et al*, 2009), and it is involved in several physiological functions such as neural development, angiogenesis, cardioprotection, prevention of oxidative stress, cell proliferation, and apoptosis (Bouillaud & Blachier, 2011). However, if accumulated in the order of micromolar concentrations, it becomes toxic through interaction with heme-containing proteins, causing severe cytochrome *c* oxidase (COX, complex IV) deficiency, through rapid *heme a* inhibition that maintains the complex in a constant reduced state and accelerates long-term degradation of COX subunits (Di Meo *et al*, 2011). In addition, sulfide inhibits the enzymatic activity of short-chain acyl-CoA dehydrogenase (SCAD), causing dicarboxylic aciduria (Pedersen *et al*, 2003), and is implicated in vasorelaxation by opening ATP-sensitive $K^+$ channels in vascular smooth muscle (Yang *et al*, 2008), inflammation (Yang *et al*, 2013), and reactive oxygen species (ROS) production (Eghbal *et al*, 2004).

$H_2S$ is catabolized in the mitochondria by the cooperation of at least four proteins, which sequentially perform the oxidation of the sulfide into a sulfate ion (Fig 1). In the first step of sulfide catabolism, sulfide-quinone reductase (SQR), an enzyme bound to the

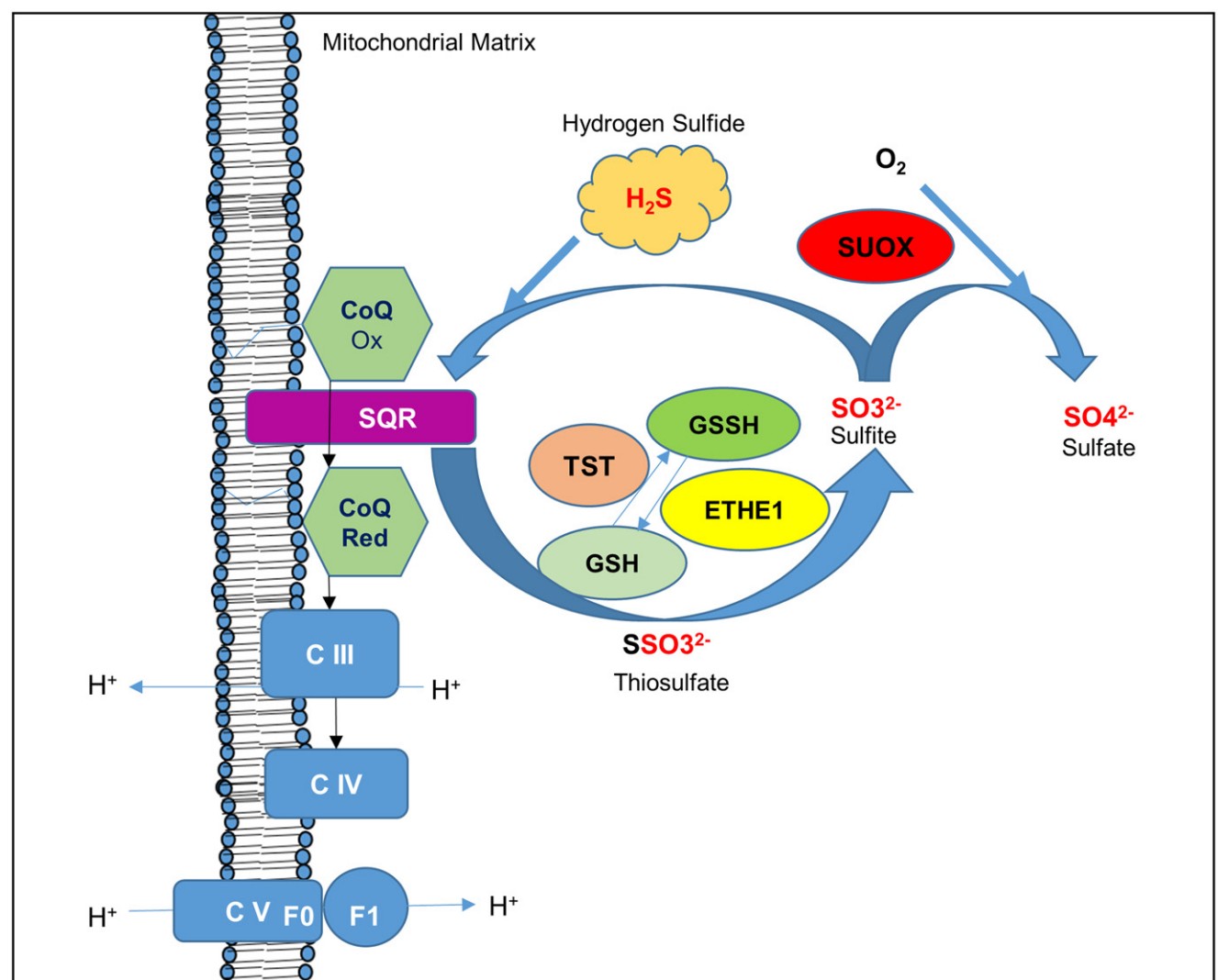

**Figure 1. Mitochondrial $H_2S$ oxidation pathway.**
SQR converts sulfide into thiosulfate by transferring two electrons from the $H_2S$ oxidation to ubiquinone (CoQ). Thiosulfate is then converted into sulfite by TST and ETHE1; this reaction requires a sulfur acceptor (glutathione, GSH). Excess of sulfite is converted into sulfate by SUOX.

    

inner mitochondrial membrane, which belongs to the family of flavoprotein disulfide oxidoreductases, transfers sulfane sulfur atoms from the toxic sulfides to free sulfites, generating thiosulfate. During this reaction, electrons are shuttled from sulfide to the mitochondrial electron transport chain by reduction of ubiquinone (CoQ) to ubiquinol (Jackson et al, 2012). The product of this reaction, thiosulfate, is then oxidized in the mitochondrial matrix by a sulfurtransferase (TST, Rodhanese). TST mediates the conversion of thiosulfate to sulfite, transferring sulfane sulfur to a –SH-containing acceptor, like glutathione (GSH). The sulfur dioxygenase ethylmalonic encephalopathy protein 1 (ETHE1 or persulfide dioxygenase), a mitochondrial matrix protein, participates in the conversion of thiosulfate to sulfite. The terminal component of this known pathway is the sulfide oxidase SUOX, which oxidizes sulfite to sulfate, which is subsequently secreted into the blood and eliminated through the urine (Muller et al, 2004; Hildebrandt & Grieshaber, 2008).

So far, molecular defects in ETHE1 and SUOX, responsible for autosomal recessive ethylmalonic encephalopathy (EE) and sulfite oxidase deficiency (or sulfocysteinuria), respectively, and the rare molybdenum cofactor deficiency, are the only cause of sulfide metabolism impairment associated with human diseases (Johnson et al, 2002; Tiranti et al, 2009).

Here, we show that in human fibroblasts, mutations in CoQ biosynthetic genes reduce SQR levels and biochemical activity proportionally to CoQ levels, followed by compensatory up-regulation of enzymes of the downstream pathway. Biochemical and molecular abnormalities are partially reverted by CoQ supplementation in vitro and recapitulated by pharmacologic inhibition of CoQ biosynthesis in wild-type skin fibroblasts and by depletion in HeLa cells of ADCK3, an atypical protein kinase involved in CoQ biosynthesis regulation (Stefely et al, 2015). In contrast, in kidneys, the only affected organ of $Pdss2^{kd/kd}$ mice, a mouse model of CoQ deficiency, severely reduced SQR levels are associated with downregulation of all the downstream enzymes of the pathway, GSH depletion, and reduction of thiosulfates. Furthermore, accumulation of $H_2S$ in $Pdss2^{kd/kd}$ mice causes short-chain CoA dehydrogenase (SCAD) inhibition, as shown by increased blood levels of C4-C6 acylcarnitines. The shutdown of the $H_2S$ oxidation pathway observed in kidneys of $Pdss2^{kd/kd}$ mice was reproduced by severe knockdown of SQR in HeLa cells. Taken together, these findings demonstrate multiple downstream effects of CoQ deficiency on sulfide metabolism.

## Results

### CoQ10 levels are variably reduced in CoQ10-deficient patient fibroblasts

To evaluate the possible relationship between CoQ10 deficiency and impairment of the hydrogen sulfide oxidation pathway, we measured CoQ10 levels in human skin fibroblasts from patients carrying molecular defects in different genes encoding proteins involved in the CoQ biosynthesis (COQ genes; Table 1 and Fig 2). Patient skin fibroblasts showed variable levels of CoQ10 deficiency ranging from 52 to 54% of normal (P1 and P5) to 28.9% (P2), and 18 and 13.4% (P3 and P4), confirming previous published findings

**Table 1.  Summary of molecular defects and CoQ10 levels in fibroblasts of patients studied.**

| Cell line | Molecular defect | Gene | CoQ10 (µg/g protein)[a] | References |
|---|---|---|---|---|
| Controls (N = 6) | | | 63.14 ± 14 | |
| P1 | Q167LfsX36 homozygous | ADCK3 | 36.9 ± 6.5 | Lagier-Tourenne et al (2008) |
| P2 | Q322X, S382L | PDSS2 | 18.3 ± 1 | Lopez et al (2006) |
| P3 | S109N homozygous | COQ2 | 11.4 ± 0.7 | Scalais et al (2013) |
| P4 | A302V homozygous | COQ2 | 8.5 ± 0.48 | Jakobs et al (2013) |
| P5 | R145G homozygous | COQ4 | 40.8 ± 5.6 | Brea-Calvo (2015) |

[a]Mean ± SD.

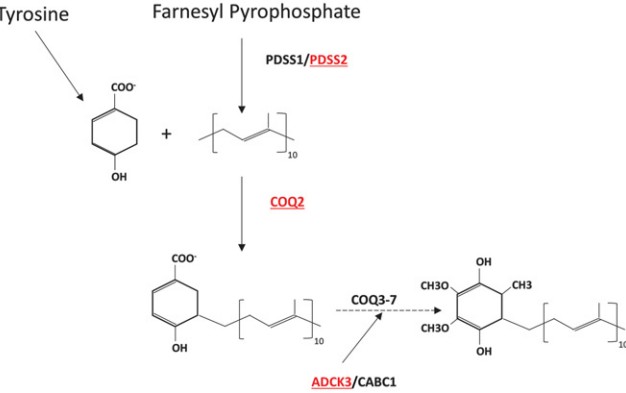

**Figure 2.  Schematic representation of CoQ biosynthesis showing the molecular defects in patient fibroblasts (P1–P5).**
The isoprene tail is synthesized from farnesyl pyrophosphate by PDSS1/PDSS2 and condensed to the 4HB ring by COQ2. The ring is processed into CoQ by COQ3-7. ADCK3/CABC1 (COQ8) regulates CoQ biosynthesis through phosphorylation of COQ3, COQ5, and COQ7.

in the same cell lines (Table 1). Intermediates of CoQ10 biosynthesis were not detected in the chromatograms of any of the patients.

Reduction in complex II-driven ADP-dependent respiration in patient fibroblasts was consistent with their levels of CoQ10 (Appendix Fig S1A and B).

### SQR activity in CoQ10-deficient patient fibroblasts is affected without compromising complex IV activity

In order to assess sulfide oxidation in primary human CoQ10 deficiencies, we evaluated oxygen consumption in the presence of NaHS, a hydrogen sulfide donor, in permeabilized human fibroblasts from patients with CoQ10 deficiency (P1–P5). Complex I inhibitor rotenone and the uncoupler FCCP were used to inhibit endogenous respiration and to ensure a maximal oxidation rate, respectively. Infusion of 60 µM NaHS allowed oxygen consumption by permeabilized cells, through the SQR-CoQ10 system, which was blocked by antimycin, an

inhibitor of the mitochondrial respiratory chain complex III. Figure 3A (solid lines) shows representative experiments of SQR-driven respiration for each cell line. In contrast, the complex IV-driven oxygen consumption rate was evaluated under the same conditions using TMPD/ascorbate, instead of NaHS, as substrates (Appendix Fig S2). The maximal respiration rate of complex IV was used to normalize sulfide oxidation values. As expected, while complex IV-driven oxygen consumption was comparable in all the tested cell lines (Appendix Fig S2), CoQ10 depletion caused a significant impairment of SQR-driven respiration in comparison with control fibroblasts, which correlated well with the residual amount of CoQ10 (Table 1). Fibroblasts from patients P2, P3, and P4 with a residual CoQ concentration below 30% showed a larger decrease in sulfide oxidation, while fibroblasts from patients P1 and P5 with a residual CoQ concentration of ~50% only showed a partial decrease in SQR-driven respiration (Fig 3A and B). Interestingly, after an initial linear decline, we observed a progressive decrease in SQR-driven oxygen consumption in patient-derived cells. This phenomenon could be explained by the reduced ability of CoQ-deficient fibroblasts to oxidize sulfide, which subsequently accumulates in the reaction media and gradually inhibits complex IV activity.

To assess whether exogenous CoQ10 can restore the ability of CoQ10-deficient fibroblasts to oxidize sulfide, fibroblasts of patients P3, P4, and P5 were cultured 1 week in the presence of 5 μM of CoQ10, before Oxygraph® analysis was performed (Fig 3A, dashed lines). While control cells did not show a significant increase in sulfide oxidation rate, COQ-mutant fibroblasts showed a significant increase in SQR-driven respiration (Fig 3B). These data confirm that CoQ10 is the electron acceptor essential for SQR activity and that human primary CoQ10 deficiency causes impairment of sulfide oxidation.

### SQR protein levels in CoQ10-deficient fibroblasts are reduced proportionally to the level of CoQ10 deficiency, and the downstream enzymes of the H₂S oxidation pathway are up-regulated

To understand whether a reduction in SQR activity in skin fibroblasts is associated with a modification of SQR protein levels, we performed Western blot in fibroblasts from patients P1 to P4. We found that SQR protein levels were significantly decreased in the two cell lines with lower CoQ10 levels (P3 and P4; 26.6 and 25.5%

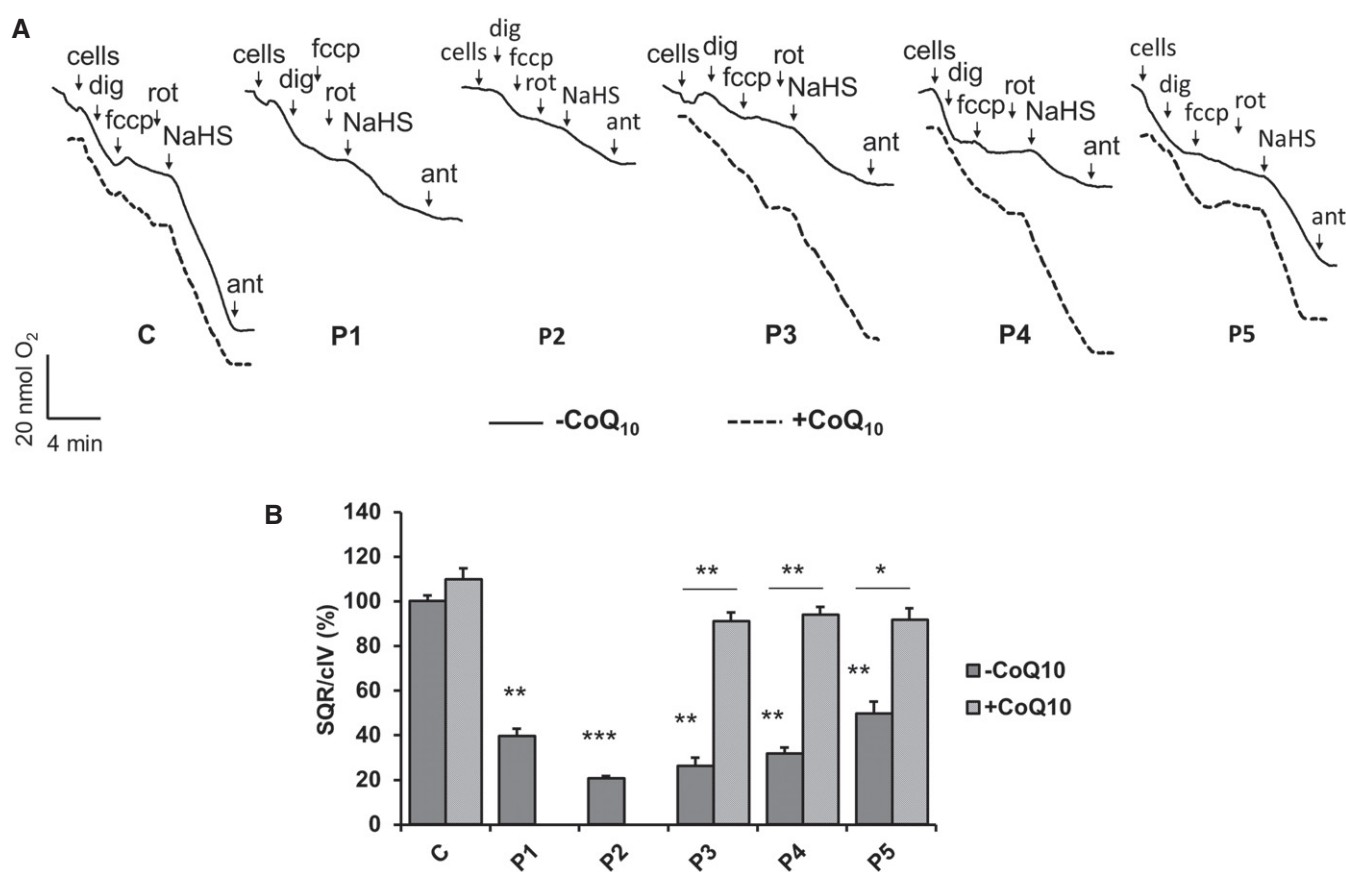

**Figure 3. Sulfide respiration in fibroblasts.**

A  Representative traces showing SQR-driven oxygen consumption in $3 \times 10^6$ control (C) and five patient fibroblast lines (P1–P5) in the absence (solid traces) or presence (dashed traces) of CoQ10 in the culture medium. See Materials and Methods for the role and concentration of the compounds used. Dig = digitonin, Rot = rotenone, fccp = carbonylcianide-p-trifluoromethoxyphenylhydrazone, Ant = antimycin, NaHS = sodium hydrosulfide.

B  Relative rates of SQR-driven oxygen consumption are normalized to CIV respiration rates. Error bars represent SDs of two experiments. Two-tailed Student's *t*-test.
   * indicates a value of $P < 0.05$, ** indicates a value of $P < 0.01$, and *** indicates a value of $P < 0.001$.

     

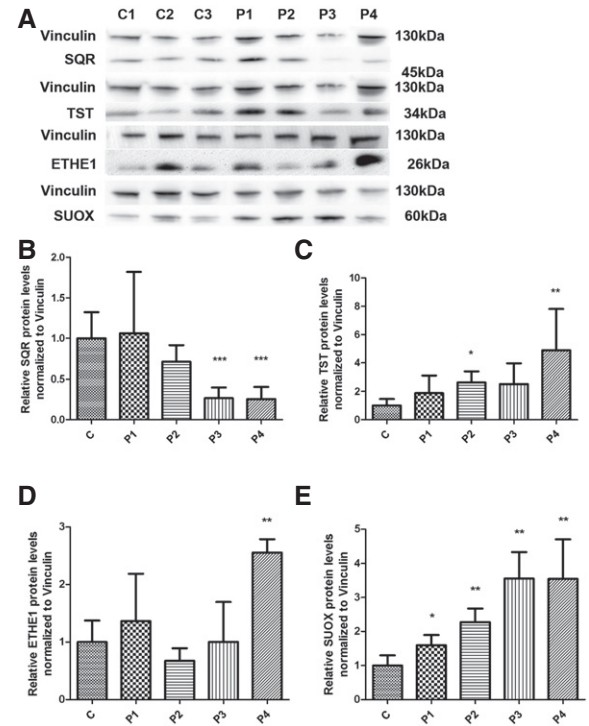

**Figure 4.  SQR, TST, ETHE1, and SUOX protein levels in fibroblasts.**

A    Representative Western blot showing the levels of SQR, TST, ETHE1, and SUOX proteins in three control (C1–C3) and four patient fibroblast lines (P1–P4).

B–E  Relative levels of proteins normalized to vinculin. Error bars represent SDs of three experiments.  Mann–Whitney $U$-test. * indicates a value of $P < 0.05$, ** indicates a value of $P < 0.01$, and *** indicates a value of $P < 0.001$.

Source data are available online for this figure.

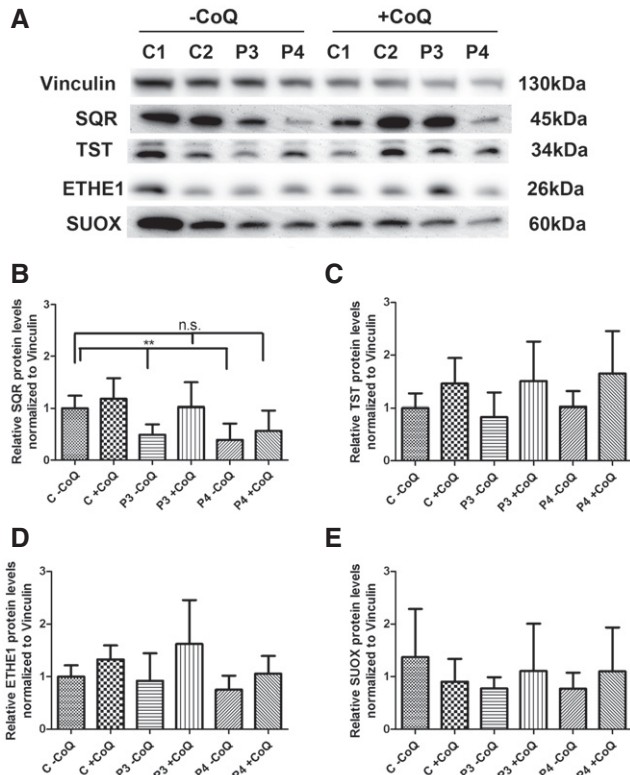

**Figure 5.  SQR, TST, ETHE1, and SUOX protein levels in fibroblasts after CoQ supplementation.**

A    Representative Western blots showing the level of SQR, TST, ETHE1, and SUOX proteins in two control (C1 and C2) and two patient fibroblast lines (P3 and P4).

B–E  Relative levels of proteins normalized to vinculin. Error bars represent SDs of three experiments. Mann–Whitney $U$-test. ** indicates a value of $P < 0.01$.

Source data are available online for this figure.

of controls; $P = 0.0003$; Fig 4A and B). SQR protein levels were also partially decreased in P2 (70.9% of controls; $P = 0.092$), but not in P1.

To investigate the consequences of reduced SQR activity/protein levels on the other mitochondrial enzymes of the $H_2S$ oxidation pathway, we measured protein levels of TST, ETHE1, and SUOX and found them to be increased in fibroblasts from patients P1–P4 (TST: 188–489% of controls; SUOX: 159–355% of controls; and ETHE1: 255% of controls in P4; for TST: $P = 0.16$, $P = 0.0115$, $P = 0.097$, $P = 0.0115$; for SUOX: $P = 0.014$, $P = 0.007$, $P = 0.007$, $P = 0.007$; for ETHE1: $P = 0.37$, $P = 0.076$, $P = 0.83$, $P = 0.002$; Fig 4C–E).

To assess whether exogenous CoQ10 was able to restore SQR protein levels, we cultured the two cell lines with lower CoQ10 and SQR levels (P3, with 49% of residual SQR, $P = 0.006$, and P4, with 39% residual SQR, $P = 0.006$) with 5 µM of CoQ10 for 1 week. We observed that CoQ10 supplementation completely restored SQR protein levels in P3 patient cell line (102% of controls, $P = 1$) and partially increased SQR protein levels in P4 (56% of controls, $P = 0.093$; Fig 5A and B). The other enzymes of the $H_2S$ oxidation pathway were unchanged by CoQ10 supplementation (Fig 5A and C–E). In summary, these data show that severe CoQ10 deficiency is associated with proportional reduction in SQR, and increased levels of the enzymes of the downstream pathway. CoQ

supplementation increases the levels of SQR, without normalizing the levels of the enzymes of the downstream pathway, suggesting that these depend on SQR levels, rather than CoQ10 levels.

**Inhibition of CoQ10 synthesis by 4-NB treatment in wild-type fibroblasts and ADCK3 knockdown in HeLa cells recapitulates the molecular phenotype of CoQ10-deficient patient fibroblasts**

To assess whether the molecular abnormalities in the $H_2S$ oxidation pathway observed in patient fibroblasts were caused by CoQ10 deficiency independently from the primary molecular defects, we used two approaches: (i) We pharmacologically inhibited CoQ10 biosynthesis in wild-type skin fibroblasts with 4-nitrobenzoate (4-NB), an analogue of 4-hydroxybenzoate (4HB) that specifically inhibits the activity of COQ2 and causes CoQ10 deficiency (Forsman et al, 2010; Quinzii et al, 2012), and (ii) we knocked down ADCK3 in HeLa cells. ADCK3 is an atypical protein kinase, whose mutations cause CoQ10 deficiency in humans and whose yeast homologue has been shown to regulate CoQ biosynthesis (Acosta et al, 2016).

Wild-type fibroblasts treated with 4-NB [4 mM] showed 40% residual CoQ10 levels (CoQ10: $19.35 \pm 3.1$ ng/mg protein in

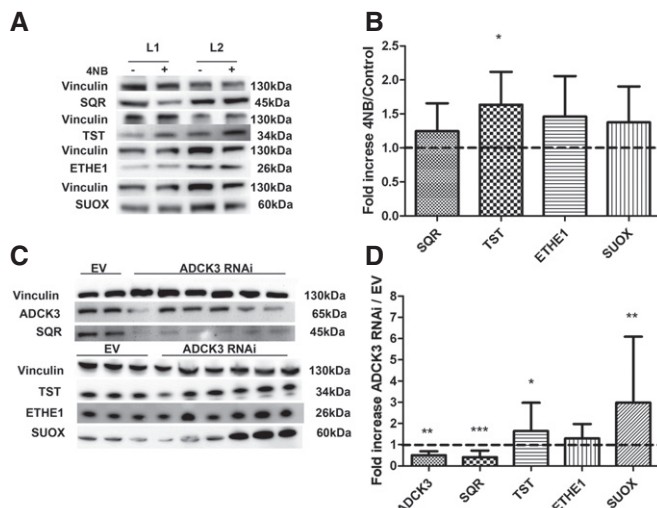

**Figure 6. SQR, TST, ETHE1, and SUOX protein levels in 4-NB-treated fibroblasts and ADCK3-depleted HeLa cells.**

A   Representative Western blots showing SQR, TST, ETHE1, and SUOX of two control fibroblast lines treated with DMSO (−) or with DMSO + 4-NB (+).
B   Quantification of protein levels in fibroblasts treated with 4-NB relative to control.
C   Representative Western blots showing ADCK3, SQR, TST, ETHE1, and SUOX in two control (EV) and six ADCK3-depleted clones (ADCK3 RNAi).
D   Quantification of protein levels in ADCK3-depleted clones (ADCK3 RNAi) relative to controls (EV).

Data information: Error bars in (B, D) represent SDs of at least three experiments with different cell lines. Paired *t*-test in (B). Mann–Whitney *U*-test in (D). * indicates a value of $P < 0.05$, ** indicates a value of $P < 0.01$, and *** indicates a value of $P < 0.001$.
Source data are available online for this figure.

4-NB-treated fibroblasts versus $50.8 \pm 14.7$ ng/mg protein of controls; $P = 0.02$), but not significant reduction in SQR (Fig 6A and B). These data indicate that (i) SQR may be degraded only in the presence of severe and constitutive CoQ10 depletion, which was not the case in 4-NB-treated fibroblasts, and (ii) transcriptional activation of *SQR* might compensate for the lack of protein activity and stability.

However, despite the normal SQR levels, we observed a significant increase in TST protein in 4-NB-treated cells (164% of controls; $P = 0.02$), as we observed in fibroblasts from patients P1 to P4, and mild, not statistically significant increase in ETHE1 and SUOX (146 and 137% of controls, respectively; Fig 6A and B).

Depletion of ADCK3 in HeLa cells (50% residual ADCK3 levels compared with controls; $P = 0.009$; Fig 6C and D) caused CoQ10 deficiency (~50% of controls; CoQ10: $149.7 \pm 31.75$ ng/mg protein in ADCK3-depleted cells versus $247.4 \pm 11.39$ ng/mg protein of control cells; $P = 0.04$) and a significant reduction in SQR protein levels (40% of controls; $P = 0.0001$; Fig 6C and D). Consistently with the observations in patient fibroblasts, in ADCK3-depleted cells, we found also increased levels of TST (164% of controls; $P = 0.038$), ETHE1 (129.5% of controls; $P = 0.36$), and SUOX (298% of controls; $P = 0.007$; Fig 6C and D).

In summary, these data confirm that reduction in CoQ10 levels, $\leq 40\%$ of physiological levels, induces SQR depletion, which is associated with up-regulation of the $H_2S$ downstream oxidation pathway.

## mRNA transcript levels of the enzymes of the $H_2S$ oxidation pathway downstream of SQR are increased in CoQ10-deficient fibroblasts

In order to understand whether changes in protein levels of the enzymes involved in $H_2S$ oxidation pathway observed in CoQ10-deficient cells are associated with modifications of mRNA transcript levels, we performed quantitative RT–PCR in fibroblasts from patients P1–P4, in wild-type fibroblasts treated with 4-NB, and in ADCK3-depleted cells.

*SQR* mRNA levels were significantly reduced in fibroblasts from P1, P3, and P4 (60, 76 and 45% of controls, respectively; $P = 0.021$, $P = 0.03$, and $P = 0.004$; Fig EV1A). However, we observed an increase in *SQR* mRNA in P2 (191%; $P = 0.021$) compared with controls (Fig EV1A). This almost twofold transcriptional up-regulation may explain why this cell line had only partially decreased SQR protein levels.

We did not find any difference in *TST* mRNA transcript levels between patients and control cells again with the exception of P2, which showed a significant increase in *TST* mRNA transcript levels (188% of controls; $P = 0.003$; Fig EV1B). *ETHE1* and *SUOX* mRNA transcript levels were significantly increased only in P3 and P4 (185% and 187 of controls, respectively; $P = 0.036$ and $P = 0.034$; Fig EV1C and D), and *SUOX* mRNA levels were increased only in P4 (190% of controls, $P = 0.004$; Fig EV1C and D).

To assess whether differences in ROS production are responsible for the alterations in mRNA expression levels of the enzymes of the $H_2S$ oxidation pathway, we performed MitoSoX staining in P3 and P4. However, fluorescence intensity was comparable between mutant and control skin fibroblasts (Appendix Fig S3). We previously excluded increased MitoSoX staining and oxidative stress in P1 and P2 cultured under the same conditions (Quinzii *et al*, 2008, 2010; Lopez *et al*, 2010).

To determine whether exogenous CoQ10 affects transcript levels of the enzymes of the $H_2S$ oxidation pathway, we measured their mRNA in P3, P4, and controls cultured with 5 µM of CoQ10 for 1 week. We observed a slight increase in *SQR* mRNA in P3 [from 75% ($P = 0.024$) to 86% of control ($P = 0.072$)], which only partly explains the increase in SQR protein level observed in the same cell line. Surprisingly, *SQR* mRNA levels were severely decreased in P4 (from 63 to 19% of control; $P = 0.0061$) and partially in control cell lines (63% of control; $P = 0.028$). In controls, also *TST* and *SUOX* mRNA levels were partially decreased (86 and 82% of untreated; $P = 0.05$ and $P = 0.007$), while *ETHE1* transcript levels were reduced in P4 (from 66 to 36% of control; $P = 0.048$ and $P = 0.004$, respectively; Fig EV2).

Inhibition of CoQ synthesis through 4-NB treatment caused a significant increase in *TST* and *SUOX* mRNA transcript levels (200 and 168% of controls; $P = 0.02$ and $P = 0.04$), and showed a trend toward an increase in *SQR* and *ETHE1* mRNA transcript levels (127 and 158% of controls; $P = 0.84$ and $P = 0.15$; Fig EV3A). These abnormalities became even more evident if the cells were supplemented with 4-NB in galactose medium (Fig EV3B).

Depletion of ADCK3 in HeLa cells caused a significant decrease in *SQR* mRNA levels (29%; $P = 0.009$ of controls; Fig EV3C). The downstream enzymes expression levels were comparable in ADCK3-depleted and control cells (Fig EV3C).

To confirm that H₂S accumulation is caused by SQR reduction due to CoQ deficiency and to exclude that H₂S increase causes reduction in SQR levels, we supplemented wild-type fibroblasts for 24 h with 0.5 mM NaHS. These conditions have been previously shown to be toxic (Di Meo *et al*, 2011). SQR mRNA and protein levels were not decreased in treated fibroblasts (Appendix Fig S4).

In conclusion, these data show that CoQ10 deficiency causes reduction in SQR at translational level. The increase in the levels of the other enzymes of the H₂S oxidation pathway appears to be a secondary response at translational level to decreased SQR levels or activity.

### Protein S-sulfhydration is increased in CoQ10-deficient cells

H₂S physiologically modifies sulfhydryl groups of some proteins, modifying their function. To prove that H₂S oxidation impairment and consequent H₂S accumulation caused by CoQ deficiency increases the levels of protein S-sulfhydration, we assessed H₂S-mediated protein sulfhydration in P1 (~50% residual CoQ; normal SQR protein levels) and P4 (~15% residual CoQ; ~26% SQR protein). As expected, we found increased protein sulfhydration in patient cell lines compared with controls (Fig 7; Appendix Table S1); however, there was no difference in the magnitudes of the increases in the two patient cell lines (1.61- and 1.49-fold increase, respectively, compared with controls; Fig 7; Appendix Table S1), indicating that in P1, although SQR protein levels are normal, the partial lack of CoQ is enough to reduce SQR activity, as shown by the experiment of SQR-driven respiration.

### Severe depletion of SQR in HeLa cells causes down-regulation of the downstream enzymes of the H₂S oxidation pathway

To further prove that the downstream enzymes of the H₂S oxidation pathway are regulated by SQR levels, we transiently knocked down SQR in HeLa cells. After 24 h of RNA interference, although *SQR* mRNA levels were severely reduced (20% of controls; $P = 0.024$), the changes in the level of SQR protein (80% of controls, $P = 0.07$) were too subtle to affect the levels of the downstream enzymes (Fig EV4A). However, after 48 h, we observed severe reduction in SQR mRNA (5% of controls; $P = 0.0003$) and protein levels (5% of controls; $P = 0.0002$; Fig EV4B, D and F), associated with down-regulation of *TST*, *ETHE1*, and *SUOX* mRNA levels (75, 52, and 76% of controls; $P = 0.1$, $P = 0.013$, and $P = 0.057$; Fig EV4B, D and F). In contrast, only TST protein levels were decreased (76% of controls; $P = 0.0005$; Fig EV4B and D).

These data indicate that reduction in SQR, below 20% of control levels, causes transcriptional down-regulation of the enzymes of the downstream H₂S oxidation pathway.

### The H₂S oxidation pathway is impaired in clinically affected tissues of *Pdss2*^kd/kd mice

To assess the relationship between CoQ levels and sulfide metabolism *in vivo*, we measured mRNA and protein levels of the enzymes involved in H₂S oxidation pathway in kidney and brain of *Pdss2*^kd/kd mice, which carry a homozygous mutation in *Pdss2*, a subunit of the first enzyme of CoQ biosynthesis (Peng *et al*, 2004). We previously demonstrated that kidney, the only clinically affected organ of these animals, had 14% residual CoQ9 (the form of CoQ more

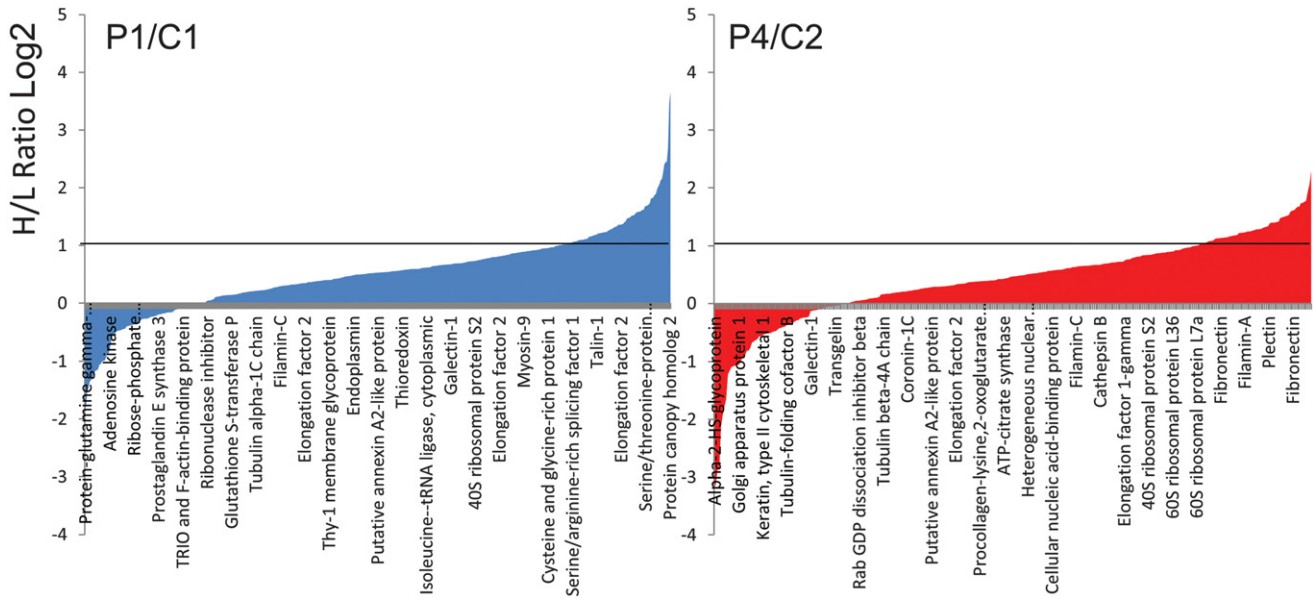

**Figure 7. Quantitative analysis of protein sulfhydration in fibroblasts.**
Distribution of peptides containing sulfhydrated cysteine residues relative to their H/L ratios as determined by the BTA analysis of cell extracts isolated from C1 and P1 patient cells (left) and C2 and P4 fibroblasts (right). Values of H/L ratios are plotted against the number of identified peptides. The gray line marks the H/L ratio > 2, consisting of cysteine-containing peptides in proteins from patient cell lines that exhibited higher reactivity with H₂S compared with the respective control. Indicated on the x-axes are representative sensitive cysteine peptides, which are common between the ones found in the RedoxDB database, and by the BTA assay.

abundant in mice) concentrations, while brain, which was clinically unaffected, had 30% residual CoQ9 concentrations (Quinzii *et al*, 2013). Consistent with our observation in mutant fibroblasts, SQR protein levels were severely reduced in kidneys of mutant (16% of controls; $P = 0.0001$) as compared with control animals (Fig 8A). In contrast to our observations in fibroblasts, which had increased TST, ETHE1, and SUOX protein levels, but consistent with the results in SQR-depleted cells, these proteins were all significantly decreased in the kidney tissue of *Pdss2*<sup>kd/kd</sup> mice (30, 34 and 45% of controls, respectively; $P = 0.0001$, $P = 0.003$, and $P = 0.003$) as compared with wild-type mice (Fig 8B–D).

SQR protein levels in brain were almost undetectable in wild-type mice (Fig 8A), while *Pdss2*<sup>kd/kd</sup> mice exhibited a trend toward increase in SQR protein levels in brain (154% of controls; $P = 0.211$; Fig 9A), which was in stark contrast to the situation in the kidney. Importantly, levels of the other enzymes of the pathway were comparable with those of control wild-type animals (Fig 9B–D).

We did not observe a reduction in *SQR* mRNA transcript levels in kidney of *Pdss2*<sup>kd/kd</sup> mice (Fig EV5A). However, as observed in patient fibroblasts (Fig EV1B) and in fibroblasts treated with 4-NB (Fig EV3A and B), mRNA transcript levels of the genes encoding the downstream pathway were increased, although only *TST* and

*ETHE1* mRNA transcript levels increased significantly (200 and 480% of controls; $P = 0.012$ and $P = 0.045$; Fig EV5A), suggesting an attempt of transcriptional up-regulation to compensate for the reduced protein levels.

### The levels of GSH are reduced in clinically affected tissues of *Pdss2*<sup>kd/kd</sup> mice

Intrigued by the observation that both TST and ETHE1 protein levels, which are involved in glutathione (GSH) regeneration, were decreased in kidney tissue of *Pdss2*<sup>kd/kd</sup> mice, we measured the total level of GSH in kidney and brain of *Pdss2*<sup>kd/kd</sup> mice. GSH levels were significantly reduced in kidney (17% of controls; $P = 0.016$), but not in brain of *Pdss2*<sup>kd/kd</sup> mice (Fig 10A and B).

### The levels of thiosulfates are reduced in urine of *Pdss2*<sup>kd/kd</sup> mice

TST transfers sulfur to GSH, thereby converting it to GSSG. The source of sulfur is thiosulfate, produced by SQR. To test the hypothesis that reduced activities of SQR and downstream enzymes would decrease

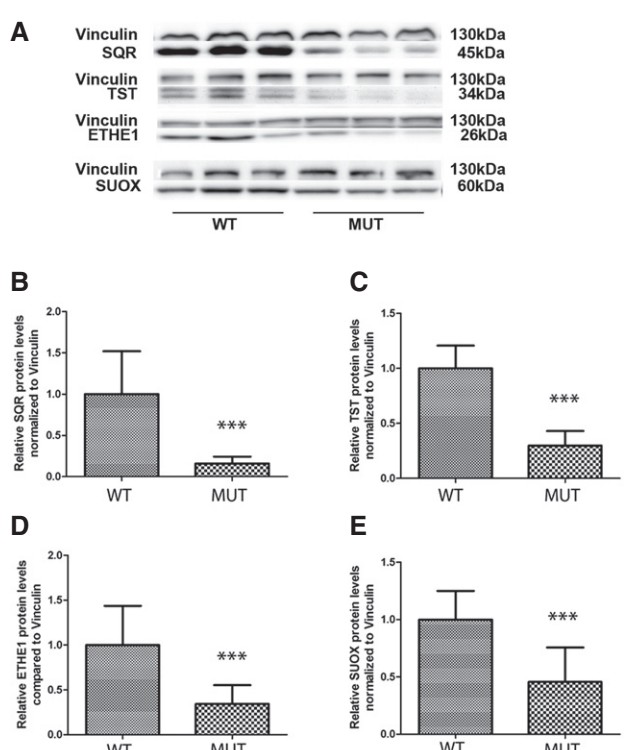

**Figure 8.   SQR, TST, ETHE1, and SUOX protein levels in the kidney of *Pdss2*<sup>kd/kd</sup> mice.**

A    Representative Western blots showing the levels of SQR, TST, ETHE1, and SUOX protein in the kidney of three wild-type (WT) and three mutant (Mut) mice.

B–E   Relative levels of proteins normalized to vinculin. Error bars represent SDs of nine animals per group. Mann–Whitney *U*-test. *** indicates a value of $P < 0.001$.

Source data are available online for this figure.

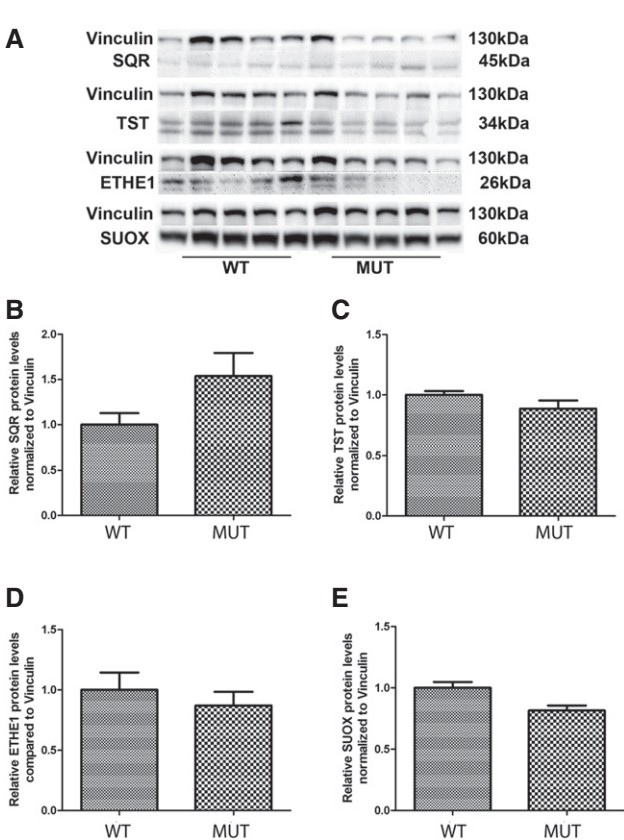

**Figure 9.   SQR, TST, ETHE1, and SUOX protein levels in brain of *Pdss2*<sup>kd/kd</sup> mice.**

A    Representative Western blots showing the levels of SQR, TST, ETHE1, and SUOX protein in the brain of five wild-type (WT) and five mutant (Mut) mice.

B–E   Relative levels of proteins normalized to vinculin. Error bars represent SDs of five animals per group. Mann–Whitney *U*-test.

Source data are available online for this figure.

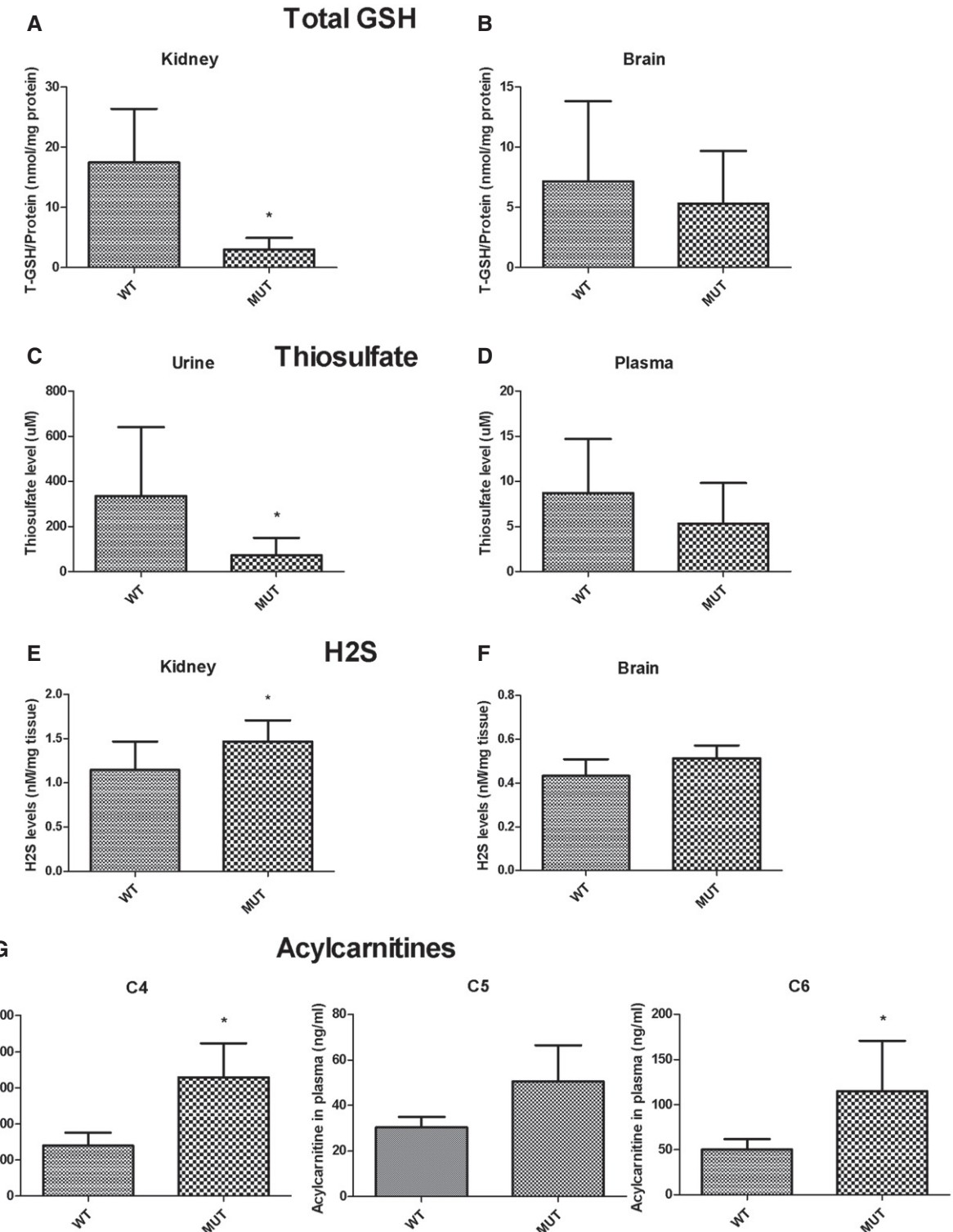

**Figure 10. GSH, thiosulfate, H₂S, and acylcarnitines levels in *Pdss2*$^{kd/kd}$ mice.**

A, B Total GSH amount normalized to protein from kidney (A) and brain (B) of wild-type (WT) and mutant (Mut) mice. Error bars represent SDs of at least three animals per group.

C, D Thiosulfate levels in urine (C) and plasma (D) normalized to protein amount. Error bars represent SDs of nine animals per group (urine) and six animals per group (plasma).

E, F H₂S amount normalized to tissue from kidney (E) and brain (F). Error bars represent SDs of seven to eight animals per group.

G Acylcarnitine C4, C5, and C6 levels from plasma, normalized to volume. Error bars represent SDs of five wild-type and three mutant mice.

Data information: Mann–Whitney *U*-test. * indicates a value of *P* < 0.05.

the production of thiosulfate, we measured thiosulfate in urine and plasma of $Pdss2^{kd/kd}$ mice. We found that the amount of thiosulfate was significantly reduced in the urine (21% of controls; $P = 0.02$), and mildly, although not significantly reduced in plasma (60% of controls) of $Pdss2^{kd/kd}$ as compared with wild-type mice (Fig 10C and D).

### Levels of $H_2S$ are increased in clinically affected tissues of $Pdss2^{kd/kd}$ mice

To investigate whether CoQ deficiency might cause an accumulation of $H_2S$ *in vivo*, we measured $H_2S$ levels in kidney and brain of $Pdss2^{kd/kd}$ and wild-type mice and found them significantly increased only in kidney of the $Pdss2^{kd/kd}$ animals (133% of controls; $P = 0.04$; Fig 10E). These data suggest that, as expected, down-regulation of the $H_2S$ catabolic pathway leads to $H_2S$ accumulation in the kidney.

### $H_2S$ accumulation in $Pdss2^{kd/kd}$ mice causes a defect of short-chain fatty acid oxidation

To assess toxic effects of $H_2S$ accumulation in $Pdss2^{kd/kd}$ mice, we determined COX activity in kidney and brain as well as the acylcarnitine profile in plasma of $Pdss2^{kd/kd}$ and wild-type mice. COX activity, normalized to protein amount and to CS and SDH activities, indices of mitochondrial mass, was not reduced in any tissue analyzed. However, C4 and C6 levels were significantly increased, and C5 was moderately increased in blood of $Pdss2^{kd/kd}$ mice (236 and 228%, and 167% of controls respectively; $P = 0.036$, $P = 0.035$, and $P = 0.14$; Fig 10G; Appendix Table S2). These data suggest that $H_2S$ accumulation causes inhibition of short-chain acyl-CoA dehydrogenase (SCAD) and a subsequent defect of short-chain fatty acid oxidation.

## Discussion

Here, we have addressed the role of an impairment of $H_2S$ oxidation in the pathogenesis of CoQ deficiency. We demonstrated that CoQ-deficient fibroblasts carrying mutations in different COQ genes have decreased SQR-driven respiration, which is associated with variably reduced SQR steady-state levels, depending on the severity of CoQ deficiency, and subsequent compensatory up-regulation of the levels of the downstream proteins of the pathway. We hypothesize that if the levels of CoQ are very low, SQR activity is decreased and SQR becomes unstable and is probably degraded. Partial reduction in SQR and/or its activity leads to up-regulation of the enzymes of the downstream pathway (TST, ETHE1, and SUOX), which is more evident at the translational than at transcriptional level. The molecular abnormalities observed in COQ-mutant fibroblasts are due to reduced CoQ levels, because they can be rescued by CoQ supplementation and are partially recapitulated by pharmacological inhibition of CoQ biosynthesis via 4-NB in wild-type fibroblasts and by ADCK3 depletion in HeLa cells. 4-NB is an analogue of 4-HB and inhibitor of 4-para-hydroxybenzoate:polyprenyl transferase (COQ2; Forsman *et al*, 2010). We previously showed that 4-NB at a concentration of 4 mM causes a CoQ deficiency of ~40% and consequent reduced ATP and ATP/ADP levels, increased ROS and oxidative stress resulting in cell death (Quinzii *et al*, 2012). Here, we found

that the same dose of 4-NB is associated with normal levels of SQR protein. We hypothesize that the decrease in CoQ levels is not severe enough to cause protein instability, or that in fibroblasts SQR is degraded only if CoQ deficiency is chronic, and that the limited timeframe of pharmacologic inhibition of CoQ biosynthesis might decrease SQR activity, but not its protein levels. Consistent with the first possibility is our observation that COQ-mutant fibroblasts with more severe CoQ deficiency (P3 and P4) have the lowest levels of SQR. P2 and fibroblasts treated with 4-NB show transcriptional up-regulation of *SQR* mRNA, not evident in the other cell lines. We cannot explain why the response of the $H_2S$ oxidation pathway to CoQ deficiency differs among patient cell lines at transcriptional levels. However, the effect of CoQ synthesis inhibition on *SQR* mRNA levels was striking, suggesting that changes in CoQ levels modulate gene expression, possibly through this antioxidant function. Indeed, a role of CoQ in several biological processes, such as lipid metabolism, inflammation, and cell signaling through regulation of genes expression, has been previously proposed (Schmelzer *et al*, 2008; Fischer *et al*, 2016). The level of the enzymes of the pathway downstream SQR appears to be regulated by SQR activity and/or level: Partial SQR depletion as observed in patient fibroblasts, 4-NB-treated wild-type fibroblasts, and ADCK3-depleted HeLa cells causes up-regulation of the other enzymes, while severe SQR depletion shuts down the pathway, suggesting a mechanism of regulation of the pathway downstream SQR possibly dependent on the availability of catabolites generated by SQR activity. Although our data suggest a tight regulation at both transcriptional and translational level, this regulatory mechanism needs further investigation.

As a consequence of sulfide accumulation, COQ-mutant fibroblasts have increased protein S-sulfhydration. Although we observed a correlation between degree of CoQ deficiency and SQR-driven respiration defect, these changes were not correlated with the magnitude of protein sulfhydration increase. This may be due to (i) experimental conditions, since SQR-driven respiration is measured adding excess of NaHS as substrate, while protein sulfhydration was measured under physiological conditions, or (ii) presence of other compensatory mechanisms regulating $H_2S$ levels, for example, effect on the upstream $H_2S$ biosynthesis pathway, depending on CoQ levels or genetic background of the fibroblasts. Regardless, CoQ deficiency clearly causes disregulation of an important post-translational modification, which may affect multiple physiological processes (Mustafa *et al*, 2009; Paul & Snyder, 2012).

The observation that NaHS supplementation in wild-type fibroblasts does not cause reduction in mRNA and protein SQR levels confirms that lack of CoQ and SQR causes sulfide accumulation, and not vice versa.

Interestingly, we found normal levels of SQR and an increase in protein levels for TST, ETHE1, and SUOX in brain of $Pdss2^{kd/kd}$ mice, which harbor a spontaneous mutation in the gene encoding the subunit 2 of polyprenyl-diphosphate synthase (*Pdss2*), the first enzyme of the CoQ biosynthetic pathway. Homozygous $Pdss2^{kd/kd}$ mice are apparently healthy for the first 8 weeks of life, but from 12 weeks of age onward, histological analysis of the kidney (which has only 15% residual CoQ) reveals a mononuclear cell infiltrate and tubular dilatation. Adult $Pdss2^{kd/kd}$ mice develop a typical nephrotic syndrome, leading to kidney failure (Peng *et al*, 2004; Madaio *et al*, 2005). In brain, which has ~30% residual CoQ concentrations and does not show any clinical phenotype, SQR was detectable only in

mutant mice, and the downstream $H_2S$ oxidation pathway was normal. It is noteworthy that we detected SQR in brain of mutant and not wild-type animals, since although *SQR* mRNA has been reported to be present in brain, the presence of SQR protein in brain is still under debate (Ackermann *et al*, 2011; Linden *et al*, 2012). In contrast, in kidney of *Pdss2*[kd/kd] mice, we did not find an up-regulation of the sulfide oxidation pathway, but rather decreases in all the enzymes of the pathway. Moreover, we observed reduced GSH in kidney and very low levels of thiosulfate in the urine of *Pdss2*[kd/kd] mice, suggesting a shutdown of the oxidation pathway, as observed in SQR-depleted HeLa cells cultured for 48 h.

As a consequence of the shutdown of the $H_2S$ catabolic pathway, we observed a mild accumulation of $H_2S$ in mutant mice kidney compared with controls. This quantification may be inaccurate, because $H_2S$ is extremely volatile and therefore difficult to measure in tissues (Vitvitsky *et al*, 2012). The only moderate increase in $H_2S$ concentrations we observed may explain the absence of COX deficiency in the affected tissues of the *Pdss2*[kd/kd] mice (Quinzii *et al*, 2012). Although COX inhibition is considered the primary mechanism of $H_2S$ toxicity (Nicholls & Kim, 1982), it is possible that in CoQ deficiency, where SQR activity is reduced but not completely absent, $H_2S$ levels are not high enough to suppress COX activity in contrast to ethylmalonic aciduria, which presents with a much more severe phenotype and COX deficiency in patients (Tiranti *et al*, 2009). Furthermore, it is important to note that the ethylmalonic aciduria mouse model shows normal COX activity and level in kidney and liver, in spite of high thiosulfate and $H_2S$ concentrations. This may reflect the presence of tissue-specific alternative metabolic pathways for $H_2S$ detoxification, or different buffering mechanisms (Tiranti *et al*, 2009). Nevertheless, other mechanisms of $H_2S$ toxicity, including inhibition of short-chain fatty acid oxidation, GSH depletion (Truong *et al*, 2006), and the ability of $H_2S$ to generate ROS (Bouillaud & Blachier, 2011), may play a role in the pathogenesis of CoQ deficiency and EE-deficient mice. We previously demonstrated increased ROS production and oxidative stress in kidney tissue of *Pdss2*[kd/kd] mice (Quinzii *et al*, 2013). The role of oxidative stress in the pathogenesis of CoQ deficiency is supported by the observation of CoQ deficiency and oxidative stress in the brain of *Coq9*-mutant mice (Garcia-Corzo *et al*, 2013). Moreover, in COQ-mutant fibroblasts, and in wild-type fibroblasts treated with 4-NB, we showed that reduced cell viability correlated with increased ROS production and oxidative stress (Lopez *et al*, 2010; Quinzii *et al*, 2010, 2012). We also observed in one COQ-mutant cell line (P2 in this work) a trend toward a decrease in total GSH content accompanied by a slight decrease in the activities of the glutathione enzymes GPx and GRd (Quinzii *et al*, 2008). Therefore, it is tempting to speculate that tissue-specific abnormalities of $H_2S$ metabolism may contribute to oxidative stress in CoQ deficiency through alteration of the glutathione system. For example, $H_2S$ autoxidation could produce reactive sulfur and oxygen radical causing GSH depletion (Truong *et al*, 2006), or synthesis of GSH could be down-regulated to balance the increase in GSH caused by decrease in TST. However, the causes of GSH reduction can be independent of $H_2S$ oxidation impairment. Since CoQ is an antioxidant, both via direct prevention of lipid peroxidation and indirect regeneration of other antioxidants such as vitamins C and E, as well as an electron carrier in the mitochondrial respiratory chain, lack of CoQ may cause an increase in ROS production and oxidative stress because antioxidant defenses

are reduced and electron transport in the respiratory chain is impaired. Therefore, chronic oxidative stress due to the lack of CoQ could be responsible for depletion of antioxidant defenses, including GSH. Importantly, Luna-Sánchez *et al* (2017) showed reduced GSH levels in brain of *Coq9*-mutant mice. However, they also observed that SQR-depleted cells have GSH levels comparable to controls, supporting the hypothesis that reduction of GSH is independent of SQR levels, or is a tissue- or cell-specific effect of low SQR (Luna-Sánchez *et al*, 2017).

We cannot discern between the single factors in the overall increase in oxidative stress in states of CoQ deficiency. ROS have been shown to have a role in the pathogenesis of nephrotic syndrome (Greenbaum *et al*, 2012; Emma *et al*, 2016); however, the relationship between oxidative stress and hydrogen sulfides in renal pathology is controversial. Although most of the literature supports the notion that hydrogen sulfide protects against oxidative stress and thus generally also against renal disease (Koning *et al*, 2015; Lobb *et al*, 2015), some recent studies suggest that treatment with the CSE inhibitor propargylglycine (PAG) inhibits endogenous $H_2S$ production and increases antioxidant enzyme expression while significantly improving renal function reducing renal damage, fibrosis, and inflammation. Another study demonstrated that PAG-treated animals exhibited significantly improved renal function and decreased renal necrosis as well as decreased ROS production, oxidative stress, and caspase activation following gentamicin treatment (Francescato *et al*, 2012).

We postulate that the discrepancy in levels of SQR between human fibroblasts and mouse kidney is mostly tissue related. Tissue specificity is typical of human and murine mitochondrial disorders, and fibroblasts are not clinically affected. We previously observed that Pdss2-mutant fibroblasts do not show the detrimental effects of CoQ deficiency observed in Pdss2-mutant mice kidney. The molecular and biochemical abnormalities observed in patient fibroblasts reflect more severe abnormalities or a selective vulnerability to the effects of CoQ deficiency of the kidney, which is often clinically affected in CoQ deficiency, independently of the molecular defect.

In conclusion, we demonstrate that impairment of hydrogen sulfide oxidation may play a synergistic role with CoQ deficiency in the pathogenesis of nephrotic syndrome, one of the most common phenotypes of human CoQ deficiency through generation of additional oxidative stress. Our results provide insight into a novel disease mechanism and into the unresolved mechanisms responsible for the efficacy of CoQ10 treatment in nephrotic syndrome (Saiki *et al*, 2008; Emma *et al*, 2016).

Furthermore, we describe for the first time a defect of fatty acid oxidation to be associated with primary CoQ deficiency. This observation deserves further confirmation in human patients, because it might explain some of the clinical features present in CoQ deficiency with a severe early-onset multi-organ phenotype, and has clear therapeutic implications.

# Materials and Methods

### Cell culture

All experiments were performed in human skin fibroblasts from at least five controls and four CoQ10-deficient patients (Table 1), unless

specified otherwise. Experiments were performed in replicate. Cells were grown in Dulbecco's minimum essential medium (DMEM) supplemented with 10% fetal bovine serum (FBS), 5 ml MEM vitamins, 5 ml MEM non-essential amino acids, 1 ml fungizone, and 5 ml penicillin–streptomycin, at 37°C in 5% $CO_2$ atmosphere.

In order to assess the effects of pharmacological inhibition of CoQ biosynthesis on sulfide metabolism, we incubated control skin fibroblasts with 4-nitrobenzoate (4-NB) for 6 days. DMEM supplemented with 4 mM 4-NB or 4 mM DMSO was added to the cells at days 1, 3, and 5, and cells were collected at day 7. To assess the effects of forcing energy production through oxidative phosphorylation on sulfide metabolism, we performed the same experiments also in glucose-free medium, by adding 4-NB to RPMI 1640 glucose-free medium supplemented with galactose, 10% regular FBS, 25 mM HEPES, 1.5 mM Glutamax, 25 mM galactose, 1 ml fungizone, and 5 ml penicillin–streptomycin (Quinzii *et al*, 2012).

To assess the effects of CoQ supplementation on the $H_2S$ oxidation pathway enzymes protein and mRNA levels, we incubated skin fibroblasts with 5 μm CoQ10 (Hydro Q Sorb Powder, Tishcon Corp., USA) for 1 week, adding fresh medium supplemented with CoQ10 four times.

In order to exclude that $H_2S$ accumulation leads to decrease in SQR levels, we supplemented wild-type fibroblasts with 0.5 mM NaHS for 24 h as previously described (Di Meo *et al*, 2011).

### Animal care

B6/*Pdss2*[kd/kd] mice were purchased from Jackson laboratory. *Pdss2*[kd/kd] mice harbor a spontaneous mutation in the gene encoding the subunit 2 of polyprenyl-diphosphate synthase (*Pdss2*) and their phenotype was previously described (Peng *et al*, 2004; Quinzii *et al*, 2013). All experiments were performed according to a protocol approved by the Institutional Animal Care and Use Committee of the Columbia University Medical Center, and were consistent with the National Institutes of Health Guide for the Care and Use of Laboratory Animals. Mice were housed and bred according to the international standard conditions, with a 12-h light/12-h dark cycle and free access to food and water.

Urine was collected at 5–6 months of age by manually restraining the mice and promptly stored at −80°C. Mutant and control animals were euthanized by rapid carbon dioxide narcosis followed by cervical dislocation at end stage of the disease (6 months). Blood was extracted from the heart and collected in tubes with EDTA. Plasma was obtained from blood by centrifugation at 4°C at 800 rcf for 15 min and kept at −80°C. Brain and kidneys were quickly removed and frozen in the liquid phase of isopentane, pre-cooled toward its freezing point (−80°C) with dry ice. Experiments were performed in 5–10 mice for each group.

### Determination of CoQ10 levels in fibroblasts

Control and mutant skin fibroblasts (P1–P4) were grown until confluence, collected and washed with 1× PBS, and the pellets were stored at −80°C. CoQ10 was isolated with the hexane/ethanol 5:2 extraction technique. The combined hexane extract was dried under slow $N_2$ gas flow and then resuspended in 1-propanol. CoQ10 levels were determined by high-performance liquid chromatography (HPLC) on a reverse-phase Symmetry® C18 3.5 mm, 4.6 × 150 mm

column (Waters), using a mobile phase consisting of methanol, ethanol, 2-propanol, acetic acid (500:500:15:15), and 50 mM sodium acetate at a flow rate of 0.9 ml/min. The electrochemical detection system consisted of an ESA Coulochem III with a guard cell (upstream of the injector) at +900 mV, conditioning cell at 600 mV (downstream of the column), followed by the analytical cell at +500 mV. The CoQ10 concentration was estimated by comparison of the peak area with those of standard solutions of known concentrations. Levels of CoQ10 in P5 were measured as described previously (Brea-Calvo *et al*, 2015).

### Oxygen consumption in fibroblasts

Oxygen consumption from $3 \times 10E6$ cells permeabilized by 80 μg/ml digitonin was monitored with a Clark-type electrode (Oxygraph Hansatech Inst., Norfolk, United Kingdom) at 37°C in a respiration buffer (pH 7.4) containing 300 mM mannitol, 10 mM KCl, 5 mM $MgCl_2$, 10 mM $KH_2PO_4$, enriched with 0.1% BSA. The maximum respiratory capacity was measured by adding 4 μM of the uncoupler carbonyl cyanide 4-(trifluoromethoxy) phenylhydrazone (FCCP). Rotenone (1 μM, complex I inhibitor), antimycin A (1 μM, complex III inhibitor), and Na-cyanide (200 μM, complex IV inhibitor) concentrations were set to reach full respiratory inhibition (data not shown). The substrate concentrations were as follows: for SQR-driven respiration: NaHS, 60 μM; for complex II-driven respiration: ADP, 1 mM and succinate, 10 mM; for complex IV-driven respiration: TMPD + ascorbate, 300 μM and 6 mM, respectively. NaHS concentration was set to prevent complex IV inhibition. All measurements were corrected by subtracting the residual oxygen consumption present after full inhibition of the respiratory chain.

In recovery experiments, cells were cultured for 1 week in the presence of 5 μM CoQ10 (Skatto, Chiesi Farmaceutici, Italy).

### Sulfide metabolism pathway studies in fibroblasts and mouse tissues

*Gene expression by real-time PCR*
Total RNA from skin fibroblasts and mouse tissues was extracted using the TRIzol method and quantified by NanoDrop®. RNA was stored at −80°C. RNA samples were digested with DNase before reverse trascription (RT). RT to obtain single-strand cDNA was executed with 0.5–1.0 μg of RNA and the Vilo2 kit (Sigma Aldrich).

To determine mRNA of the *SQRDL, TST, ETHE1,* and *SUOX* genes in cells and in mouse tissues, quantitative RT–PCR was performed using TaqMan® Assays with the following Applied Biosystems probes: *SQRDL*, Hs01126963_m1 for humans, Mm00502443_m1 for mouse; *TST*, Hs00361812_m1 humans, Mm01195231_m1 for mouse; *SUOX*, Hs04183429_m1 for humans, Mm00620388_g1 for mouse; *ETHE1*, Hs00204752_m1 for humans, Mm00480916_m1 for mouse. Expression of the target genes was calculated by $-\Delta\Delta C_T$ method and normalized to the expression of *GAPDH* (#402869 for humans, #4308313 for mouse). The experiments were executed in technical triplicates and repeated with different biological replicates at least three times.

*Determination of protein steady-state levels by immunoblotting*
To measure the steady-state protein levels of the enzymes known to be involved in sulfide metabolism, we performed Western blots.

Proteins were extracted from cell pellets by sonication and from mouse tissues by mechanical homogenization in lysis buffer (50 nM Tris–HCl, 150 mM NaCl, 1 mM EDTA). To prevent protein degradation, a protease inhibitor cocktail (Complete Mini®, EDTA-free, 11836170001, Roche) was added to the protein extract and the samples were kept at −80°C. Cell lysates were quantitated for total protein content using the Bradford system (ThermoScience) and analyzed by electrophoresis in a 12–15% PAGE gel or Novex 10–20% Glycine Gel (EC61355, Invitrogen) loading 10/40 μg of protein for sample. After electrophoresis, proteins were transferred to a PVC transfer membrane (IPFL00010, Immobilon-FL). Membranes were blocked in PBT with 2–5% milk or 2% BSA before incubation with the following antibodies: rabbit anti-SQR (1:1,000, ab118772 Abcam for mouse; 1:1,000, 17256-1-AP Proteintech for humans); rabbit anti-TST (1:1,000, ab155320, Abcam for mouse; 1:1,000, 16311-1-AP Proteintech for humans); mouse anti-SUOX (1:1,000, ab57852, Abcam); rabbit anti-ETHE1 (1:1,000, Abcam ab154041); rabbit anti-ADCK3/CABC1 (1:1,000, Thermo Scientific, Pierce, PA5-13906) mouse anti-vinculin (1:5,000, Abcam ab18058) for both mouse and humans; secondary rabbit and mouse HRP (1:2,000, Sigma A9044 and A0545). Protein bands were visualized by chemiluminescence, using ECL reagents (GE Healthcare). Intensity of the bands was quantified with ImageJ (NIH).

### Generation of cells expressing shRNA

HeLa cells were cultured in DMEM with 10% FBS until 70–80% confluence. Transfections with scramble shRNA-pLKO plasmid (used as control) or ADCK3-specific TRC shRNA-pLKO plasmid construct (SHCLNG-NM_020247 MISSION® shRNA Bacterial Glycerol Stock; Sigma Aldrich) were mediated by Lipofectamine 2000 (Invitrogen) according to the manufacturer's instructions. After 5 h of transfection, cells were selected with puromycin/neomycin in DMEM 2% FBS and transfected clones expanded separately with DMEM 10% FBS.

Transient knockdown of *SQR* in HeLa cells was obtained incubating cells for 24 and 48 h in OptiMEM medium supplemented with 5 ml MEM vitamins and 5 ml MEM non-essential amino acids, 25 pmol of SQRDL Silencer Select Validated siRNA (Ambion 4390824), and Lipofectamine 2000.

### Protein sulfhydration studies

Fibroblasts of two different control lines (C1 and C2) and two different patient cell lines (P1 and P4) were grown in duplicate, and individually, lysate was then clarified by centrifugation at 4°C. Protein concentrations were determined by the BCA assay. Equal amount (1.2 mg) of proteins from each sample was incubated with 200 μM NM-biotin for 30 min with occasionally gentle mixing at room temperature, and subsequently precipitated by cold acetone. After centrifugation, pellets of precipitated proteins from each sample were washed individually with the cold acetone twice and then suspended in denaturation buffer containing urea. The denatured, biotinylated proteins were then diluted with 10 volumes of the suspension buffer and incubated with modified porcine trypsin with mixing for 18 h at 30°C. After the digestion, trypsin was inactivated by incubation at 95°C for 10 min; then, four reaction mixtures (C1, C2, P1, and P4) were mixed separately with the

streptavidin-conjugated agarose beads and incubated at 4°C for 18 h following extensive washes in the presence of low concentration of SDS. Peptides from each column were eluted individually with reducing agent after 25-min incubation at room temperature. Free -SH groups from two control fibroblasts (C1 and C2) were alkylated separately by NEM (light), whereas deuterium NEM (heavy) was used to label the patient samples (P1 and P4). After the labeling step, excess of DTT was supplemented into the reaction mixtures, then incubated for 25 min at room temperature. The labeled peptides from the C1 and P1 were combined to form one pair sample, whereas the C2 was pooled with P4. Each pair peptide sample was concentrated and desalted with a C-18 column for LC-MS/MS analysis. The detailed descriptions of peptide identification and quantification by LC/MS were carried out as described in Gao *et al* (2015).

### GSH measurement in mouse tissues

The level of total GSH (T-GSH) was measured according to the protocol described in Quinzii *et al* (2008). Briefly, frozen tissue homogenates were resuspended in 1,000 μl of 6% metaphosphoric acid, vortexed, incubated at room temperature for 10 min, and centrifuged at 15,700 *g* for 10 min at 4°C. Pellets were stored at −80°C for protein determination, and the supernatants were used to measure GSH. One hundred μl of sample, 750 μl of phosphate buffer, 50 μl of 10 mM DTNB, and 80 μl of 5 mM NADPH were added to a 1-ml cuvette. After a 3-min equilibration period at 30°C, 20 μl (1.35 U) of GRd was added, and the change in absorption over time was measured for additional 3 min. Absolute concentrations were determined using a standard curve of GSH (from 0.5 to 20 μg/ml prepared in 6% metaphosphoric acid and diluted in phosphate buffer). T-GSH concentration was expressed in nmol/mg protein.

### Thiosulfate measurements in mice

Thiosulfate concentration was measured in plasma and urine collected from mice as previously described (Völkel & Grieshaber, 1992). Briefly, 2.9 mM monobromobimane was added to both urine and serum samples. The fluorescent products were separated by HPLC with a gradient pump and an increasing gradient of methanol, using a LiChrospher 60 RP-select B, 125-4 (5-μm) column (MERCK, Darmstadt, Germany). Fluorescence emission at 480 nm was detected online after excitation at 380 nm.

### $H_2S$ measurement in mouse tissues

$H_2S$ levels were measured by precipitation of $H_2S$ with zinc acetate followed by quantitation with methylene blue as previously described (Gilboa-Garber, 1971; Ang *et al*, 2012).

### Statistical analysis

In order to compare results of CoQ, thiosulfate, GSH, and acylcarnitine measurement, Western blots, and quantitative PCR in cells and animal tissues, the Mann–Whitney non-parametric *U*-test was used, unless specified otherwise. For fibroblast 4-NB supplementation studies, a paired *t*-test was used. To compare determination of SQR-driven respiration and complex IV activity in fibroblasts, a

**The paper explained**

**Problem**

Different pathomechanisms may explain the clinical heterogeneity of CoQ deficiency. The role of sulfide oxidation impairment has never been addressed in CoQ deficiency.

**Results**

Mutant skin fibroblasts from patients with CoQ deficiency have reduced SQR-driven respiration and SQR levels, which correlate with the level of CoQ. Biochemical and molecular abnormalities are partially rescued by CoQ supplementation and recapitulated by down-regulation of CoQ biosynthesis in fibroblasts and in HeLa cells. Kidneys, the clinically affected organ of the CoQ-deficient Pdss2-mutant mice, show reduction in the levels of SQR, and of the other enzymes of the $H_2S$ oxidation pathway, accumulation of $H_2S$, and reduced total glutathione levels. Pdss2-mutant mice have also reduced urine thiosulfate levels and plasma accumulation of C4-C6 acylcarnitines.

**Impact**

Our findings provide the first evidence of a defect of $H_2S$ oxidation and consequent defect of short-chain fatty acid oxidation in CoQ deficiency, which may have important therapeutic implications.

two-tailed Student's *t*-test was used. For statistical analysis, GraphPad Prism v5 and Microsoft Excel were used. Data are expressed as mean $\pm$ SD of at least three experiments for group. A value of $P < 0.05$ was considered to be statistically significant. * indicates a value of $P < 0.05$, ** indicates a value of $P < 0.01$, and *** indicates a value of $P < 0.001$.

**Expanded View** for this article is available online.

## Acknowledgements

We acknowledge the Cell Lines and DNA Bank of Pediatric Movement Disorders and Neurodegenerative Diseases of the Telethon Network of Genetic Biobanks (Grant GTB12001J) and the Eurobiobank Network, and Prof. Michel Koenig for providing mutant cell lines. This work was supported by NIH P01 HD080642-01 (CMQ), NIH Clinical Translational Science Award (CTSA) UL1TR000040, NIH R37-DK060596 and R01-DK053307 (MH), and by the Pierfranco e Luisa Mariani Foundation, Milan (Italy). CMQ receives funding also by the Muscle Dystrophy Association (MDA) and the Department of Defense (DOD). IDM is supported by the Italian Ministry of Health Grant 79/GR-2010-2306756. The Orbitrap Elite instrument was purchased with NIH shared instrument grant 1S10RR031537-01 (BW).

## Author contributions

MZ performed the majority of the *in vitro* experiments and wrote the manuscript; IDM performed polarography and thiosulfate analysis in mouse urine and blood and wrote the manuscript; GK performed the majority of the mouse experiments and wrote the manuscript; X-HG performed protein sulfhydration studies; EB and MJS-Q contributed to some of the *in vitro* and *in vivo* experiments; ST measured CoQ by HPLC; HJ and CQ measured acylcarnitines in mice blood; RJR, ES, and MS diagnosed and followed the patients and provided mutant skin fibroblasts for the study; BW performed the proteomic analysis with LC-MS/MS; MH supervised protein sulfhydration studies. CMQ and VT designed the study, supervised the experiments, and wrote the manuscript. All authors critically reviewed the manuscript.

## Conflict of interest

The authors declare that they have no conflict of interest.

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
