## [Review Process File · EMBO Molecular Medicine]

Coenzyme Q deficiency causes impairment of the sulfide oxidation pathway

Marcello Ziosi, Ivano Di Meo, Giulio Kleiner, Xing-Huang Gao, Emanuele Barca, Maria J Sanchez-Quintero, Saba Tadesse, Hongfeng Jiang, Changhong Qiao, Richard J Rodenburg, Emmanuel Scalais, Markus Schuelke, Belinda Willard, Maria Hatzoglou, Valeria Tiranti and Catarina M Quinzii

Corresponding author: Catarina Quinzii, Columbia University Medical Center

Review timeline:

Submission date:	29 February 2016
Editorial Decision:	14 April 2016
Revision received:	15 September 2016
Accepted:	19 October 2016

Transaction Report:

Editor: Roberto Buccione

1st Editorial Decision

14 April 2016

Thank you for the submission of your manuscript to EMBO Molecular Medicine. We are sorry that it has taken longer than usual to get back to you on your manuscript. In this case we experienced some difficulties in securing three appropriate expert reviewers, also due to the request to review two back-to-back submissions, and then obtaining their evaluations in a timely manner. Furthermore, one reviewer (#1) ultimately did not deliver.

As you will see the two Reviewers are globally positive, but do raise many issues. Reviewer 3, especially, raises an important and fundamental one. Although I will not dwell into much detail, I would like to highlight the main points.

Reviewer 2 raises a number of concerns that require your action. For instance s/he notes the lack of correlation between residual CoQ levels and sulfide oxidation and would like to better understand why GSH levels are decreased although both SQR and TST are diminished in the Pdss2 mutant kidneys. The reviewer also notes that causality between low SQR and up-regulation of downstream enzymes is not established. S/he also list additional points focused on improving precision and quality of controls

Reviewer 3 feels that, in addition to other items of concern, without improved mechanistic understanding and more conclusive demonstration of causal links, the manuscript would not be suited for publication. Specifically, s/he would like to understand how SQR activity is suppressed by decreased CoQ and also raises the same concern as Reviewer 2 on the relation between CoQ levels and sulphydration. I fully agree that your work should thus be further developed in a mechanistic

sense. I should also mention that when deciding whether to send your manuscript out for review, I had sought counsel from an external advisor who agreed that the manuscript (s) was very interesting but noted that the potential shortcoming that the mechanisms were not clearly defined.

In conclusion, while publication of the manuscript cannot be considered at this stage, given the potential interest of your findings, we have decided to give you the opportunity to address the above concerns. We are thus prepared to consider a substantially revised submission, with the understanding that the Reviewers' concerns must be addressed with additional experimentation as appropriate and that acceptance of the manuscript will entail a second round of review.

I look forward to seeing a revised form of your manuscript as soon as possible.

***** Reviewer's comments *****

Referee #2 (Comments on Novelty/Model System):

Use of both patients' fibroblasts and organs from mutated mouse is appreciated

Referee #2 (Remarks):

The authors have investigated the effects of Coenzyme Q deficiency on sulfide oxidation that uses CoQ as acceptor of sulfide CoQ oxidoreductase (SQR), linking it to the respiratory chain. The study has been performed both in human fibroblasts from patients with different mutations in CoQ biosynthesis and in a mouse model with a mutation in *Pdss2*, a subunit of the first enzyme of the biosynthetic pathway of CoQ. In the fibroblasts as well as in the kidneys of the mutant mice the SQR activity and protein levels were decreased. On the other hand, the levels of downstream enzymes of the sulfide oxidation pathway generally increased in fibroblasts but decreased in mice. The kidneys of the mutant mice exhibited higher sulfide levels and decreased glutathione. In addition the mutant mice had high levels of short-chain acyl-carnitines probably caused by sulfide inhibition of their oxidation, although the sulfide levels were not enough to inhibit cytochrome oxidase.

This is a careful study considering how several aspects of sulfide metabolism may be affected by CoQ deficiency. The study has implications on the understanding some clinical features of CoQ deficiency in humans. Although the manuscript is clear and well organized, there are some points that need correction or clarification.

1. Introduction. Not all abbreviations are defined (for example ETHE1 and SUOX); the activities of the enzymes should be better defined, e.g. ETHE1 as a dioxygenase
2. Introduction and Fig. 1. The sulfide oxidation pathway is not defined clearly. For example in Fig. 1 TST and not SQR appears to be involved in formation of thiosulfate; on the other hand, in the text (line 118) it is stated that ETHE1 is involved in the conversion of sulfite to sulfate and not of thiosulfate to sulfite.
3. Results, oxygen consumption by sulfide. To better identify the level of CoQ deficiency, what are the corresponding activities for succinate oxidation in the different cells?
4. Table 1 and Fig. 3. There is no strict correspondence between residual CoQ levels and sulfide oxidation activity. For example P1 has the highest level among patients, but activity is lower than in P5; P2 has higher levels than P3, P4, P5 but has the lowest activity. Can this be explained?
5. Line 176. Can we state with certainty that it is the low SQR protein level to cause up-regulation of the downstream enzymes?
6. Lines 189-192. The data are not shown. Is recovery of SQR complete?
7. Lines 236-238. Can the authors suggest some reason for the variable effect on transcript levels in the different samples? Can different levels of CoQ biosynthetic intermediates or different extents of

ROS be a reason?

8. Line 249. Quinzii et al 2012 does not seem to be the correct reference for the CoQ levels in mutant mice.

9. Lines 269-273. I do not understand the reasoning: if SQR and TST are both decreased in kidney of Pdss2 mutants, then there should be less GSH used; therefore it seems unclear why GSH levels are decreased. The reason might be a different one from that proposed by the authors.

10. Line 444. Could intermediates of CoQ10 biosynthesis be detected in the chromatograms of some patients? Have the levels found in previous studies of CoQ (CoQ9 and CoQ10) been confirmed in this study?

11. Line 537 and following. Succinate dehydrogenase activity is mentioned nowhere in the text.

Referee #3 (Comments on Novelty/Model System):

The manuscript presents novel findings, but is largely phenomenological, failing to address molecular mechanisms that provide an explanation for changing levels of sulfide oxidation pathway enzymes when CoQ10 is deficient.

Referee #3 (Remarks):

This manuscript reports that aberrantly low Coenzyme Q levels, arising from mutations in CoQ biosynthetic enzymes, is associated with impaired hydrogen sulfide oxidation and altered expression of mitochondrial sulfide oxidation pathway enzymes - including a suppressed level of first enzyme in this pathway, SQR. In a study of skin fibroblasts from patients with attenuated CoQ levels, Ziosi et al demonstrate an impaired oxidation of hydrogen sulfide that can be rescued by CoQ supplementation. Further, in a mouse genetic model associated with low CoQ levels, the authors show diminished SQR abundance, sulfide accumulation, glutathione depletion and kidney failure.

Major Points

1. Increased protein sulphydration, owing to failure of H₂S oxidation, is presumed by the authors to be a major basis for pathology associated with CoQ-depleting gene mutations. Experiments should be performed to determine whether CoQ levels are indeed inversely correlated with the extent of protein sulphydration. Thus, it is recommended that the authors quantify the extent of protein sulphydration in genetically CoQ deficient patient cell lines (using any of several reported proteomic methods) and determine the extent to which CoQ levels are negatively correlated with total protein sulphydration and/or sulphydration of specific proteins.

2. The authors report that SQR protein levels are reduced in proportion to CoQ10 levels in patient fibroblasts. Some consideration of the mechanistic basis for this phenomenon is needed. Does SQR loss arise from decreased SQR gene transcription, SQR protein translation, or accelerated SQR protein turnover? Is SQR loss driven by accumulation of sulfide in cells, i.e., can the effect be mimicked by chronic sulfide exposure of cells possessing control levels of CoQ10?

3. The authors demonstrate that exogenous CoQ10 can restore the ability of CoQ-deficient mutant cells to oxidize sulfide (Fig 3B). Surprisingly, this finding is made after a one-week exposure to CoQ10. While it is inferred that the observed restoration of sulfide oxidation is due to CoQ10 repletion, another possibility is that SQR levels are restored by this long duration of CoQ10 treatment. It is recommended that SQR levels are quantified in CoQ10-treated cells to ascertain whether SQR abundance is restored by CoQ10 treatment. Notably, it's surprising that CoQ10 repletion would require one week of incubation. Is this long incubation period necessary to replete cellular levels of CoQ10?

4. Fig 4: While CoQ and SQR levels decrease to the same degree in both P3 and P4 cell lines, TST and ETHE1 protein levels are only significantly increased in the P4 cell line. Can the authors provide an explanation for this apparent inconsistency? Was P5 purposefully omitted in this figure.

5. Fig 7A: In both the Results and Discussion sections, the authors state that SQR was almost undetectable in brains of WT mice and the level highly increased in mutant animals. This is not apparent in the presented data. Indeed, SQR protein bands seem to be present with similar intensities. Also a p-value has not been indicated on the graph - is the observed difference statistically significant? What is the basis for normalization of western blot findings to vinculin.

6. The discrepancy between findings made with human fibroblasts and the mouse model of CoQ10 deficiency is not been adequately addressed in the Discussion section. Is it organism or tissue related? Please comment.

Minor Points

Table I: The CoQ level for P5 has been omitted from the table, presumably by accident. Please add this value.

Fig 1. Labels for sulfide oxidation enzymes are provided as a black text on a dark background - barely legible. Please modify to improve legibility.

Fig 5: On line 201 the text refers to Fig 5C, D. This should be corrected as 5A, B. References to supplementary figures between Lines 260-266 need to be corrected. (Figure S3 apparently refers to S4, S2 refers to S3 and S1 refers to S2)

1st Revision - authors' response

15 September 2016

Referee #2 (Remarks):

"This is a careful study considering how several aspects of sulfide metabolism may be affected by CoQ deficiency. The study has implications on the understanding some clinical feature of CoQ deficiency in humans. Although the manuscript is clear and well organized, there are some points that need corrections or clarification"

We are grateful to the reviewer for the positive comments on our study.

1. Introduction. Not all abbreviations are defined (for example ETHE1 and SUOX); the activities of the enzymes should be better defined, e.g. ETHE1 as a dioxygenase

We defined the abbreviations and the activities of these enzymes as follow:” Then, the sulfur dioxygenase ethylmalonic encephalopathy protein 1 (ETHE1 or persulfide dioxygenase), a mitochondrial matrix protein, participates at the conversion of thiosulfate to sulfite. The terminal component of this known pathway is the sulfide oxidase SUOX, which oxidizes sulfite to sulfate, which is subsequently secreted into the blood and eliminated through the urine (Muller et al. 2004, Hildebrandt and Grieshaber 2008)”

2. Introduction and Fig.1. The sulfide oxidation pathway is not defined clearly. For example in Fig. 1 TST and not SQR appears to be involved in formation of thiosulfate; on the other hand, in the text (line 118) it is stated that ETHE1 is involved in the conversion of sulfite to sulfate and not of thiosulfate to sulfite.

We apologize for the mistakes. In fact, the order of the enzymes of the sulfide oxidation pathway is controversial. We believe that SQR converts sulfite into thiosulfate by transferring a sulfur group from H₂S to thiosulfate. The reaction requires the reduction of ubiquinone (CoQ). Thiosulfate is then converted into sulfite by TST and ETHE1; this reaction requires a sulfur acceptor (glutathione, GSH). Excess sulfite is converted into sulfate by SUOX. We corrected Fig.1 to match the text.

3. Results, oxygen consumption by sulfide. To better identify the level of CoQ deficiency, what are the corresponding activities for succinate oxidation in the different cells?

Succinate oxidation in patient cells is reduced proportionally to their CoQ levels. The data have been added to the Results section and Appendix Fig. S1

4. Table 1 and Fig. 3. There is no strict correspondence between residual CoQ levels and sulfide oxidation activity. For example P1 has the highest level among patients, but activity is lower than in P5; P2 has higher levels than P3, P4, P5 but has the lowest activity. Can this be explained?

We agree that the correspondence between levels of CoQ and SQR driven respiration is not strict. We hypothesize that the impairment of respiration with NaHS as substrate correlates with a certain range of CoQ deficiency. Here we have shown that >50% residual CoQ is associated with milder defect of SQR driven respiration compared with <50% (14%-29%) residual CoQ, which is associated with severe defect of SQR driven respiration. This result is consistent with our previous observations in fibroblasts with different degree of CoQ deficiency: >50% (51%-69%) residual CoQ is not associated with defect in ATP synthesis, while <50% (12%-42%) residual CoQ is associated with decreased levels of ATP and ATP/ADP (Quinzii et al., 2010; Lopez et al., 2010; Quinzii et al., 2012).

5. Line 176. Can we state with certainty that it is the low SQR protein level to cause up-regulation of the downstream enzymes?

We postulate that SQR activity and/or levels determine the levels of the downstream pathway enzymes. Up-regulation of the downstream enzymes compensates for the low levels of SQR activity, or SQR protein levels, when this is >20% residual levels. However, severe reduction of SQR levels (<20% residual SQR) is associated with reduction of all the enzymes of the downstream pathway. These conclusions are based on the following results:

- 1) Patients fibroblasts with CoQ deficiency have reduced SQR activity and increased downstream enzymes levels, independently of SQR protein levels (Fig.4)
- 2) CoQ supplementation in fibroblasts increases SQR levels in patients cell lines, while the other enzymes levels are unchanged (Fig.5)
- 3) In Hela cells, depletion of the CoQ biosynthesis regulatory protein ADCK3 causes reduction of SQR (40% residual), and increase of the downstream enzymes levels (Fig.6).
- 4) Knock down of SQR (5% residual levels) in Hela cells (Fig.EV4 A, B, C) causes down-regulation of *TST*, *ETHE1*, and *SUOX* mRNA levels (Fig. EV4C).
- 5) Kidney of *Pdss2* mice show 16% residual levels of SQR, and reduction of the levels of all the other enzymes (Fig. 8)

6. Lines 189-192. The data are not shown. Is recovery of SQR complete?

CoQ₁₀ supplementation in the two patients cell lines with more severe CoQ deficiency significantly increased SQR protein levels in P3 and partially in P4 (Fig.5)

7. Lines 236-238. Can the authors suggest some reason for the variable effect on transcript levels in the different samples? Can different levels of CoQ biosynthetic intermediates or different extents of ROS be a reason?

We measured CoQ levels by HPLC and we did not observe CoQ biosynthetic intermediates in any samples. We also assessed ROS production by MitoSox, a fluorescent probe specific for mitochondrial O₂⁻, in mutant fibroblasts and we did not find any differences among mutant cell lines or between mutant cell lines and controls (Appendix Fig S3).

Our data suggest a variable level of transcriptional up-regulation of the genes encoding enzymes of the H₂S oxidation pathway downstream of SQR in COQ mutant fibroblasts, possibly related to the genetic background. However, changes in CoQ levels, induced by CoQ synthesis inhibition, clearly affect H₂S oxidation enzymes gene expression, possibly through his antioxidant function. Indeed, a role of CoQ on several biological processes, such as lipid metabolism, inflammation, and cell signaling through regulation of genes expression has been previously proposed (Fisher 2015; Schmelzer 2008)

8. Line 249. *Quinzii et al 2012 does not seem to be the correct reference for the CoQ levels in mutant mice.*

We corrected the reference, which is Quinzii et al., 2013

9. Lines 269-273. *I do not understand the reasoning: if SQR and TST are both decreased in kidney of Pdss2 mutants, then there should be less GSH used; therefore it seems unclear why GSH levels are decreased. The reason might be a different one from that proposed by the authors.*

We agree that there are different possible explanations. To address these possibilities, we have added the following paragraph to the Discussion: “Therefore, it is tempting to speculate that tissue-specific abnormalities of H₂S metabolism may contribute to oxidative stress in CoQ deficiency through alteration of the glutathione system. For example, H₂S autoxidation could produce reactive sulfur and oxygen radical causing GSH depletion (Truong et al), or synthesis of GSH could be down-regulated to balance the increase of GSH caused by decrease of TST. However, the causes of GSH can be independent of H₂S oxidation impairment. Since CoQ is an antioxidant, both via direct prevention of lipid peroxidation and indirect regeneration of other antioxidants such as vitamins C and E, as well as an electron carrier in the mitochondrial respiratory chain, lack of CoQ may cause an increase in ROS production and oxidative stress because antioxidant defenses are reduced and electron transport in the respiratory chain is impaired. Therefore, chronic oxidative stress due to lack of CoQ could be responsible for depletion of antioxidant defenses, including GSH. Importantly, Luna-Sanchez and colleagues showed reduced GSH levels in brain of *Coq9* mutant mice. However, they also observed that SQR depleted cells have GSH levels comparable to controls (Luna-Sanchez, co-submitted), supporting the hypothesis that reduction of GSH is independent of SQR levels or it is tissue-specific”.

10. Line 444. *Could intermediates of CoQ10 biosynthesis be detected in the chromatograms of some patients? Have the levels found in previous studies of CoQ (CoQ9 and CoQ10) been confirmed in this study?*

We did not detect CoQ intermediates in any samples, by HPLC. In this study we confirmed the levels of CoQ found in previous studies in all cell lines but P5, which was used only in the oxygen consumption experiment, and was not available for other experiments.

11. Line 537 and following. *Succinate dehydrogenase activity is mentioned nowhere in the text.*

We apologized for not explaining that succinate dehydrogenase activity was used as a marker of mitochondrial mass. In the Results section the following sentence “We did not detect a COX deficiency in any tissue analyzed” was changed in to “COX activity, normalized to protein amount and CS activity or SDH activity, indices of mitochondrial mass, was not reduced in any tissue analyzed”.

Referee #3 (Remarks):

The manuscript presents novel findings, but it is largely phenomenological, failing to address molecular mechanisms that provide an explanation for changing levels of sulfide oxidation pathway enzymes when CoQ10 is deficient

We thank the reviewer for recognizing the novelty of our results.

To explain the molecular mechanisms underlying the changes of the H₂S oxidation pathway enzymes observed in CoQ deficient patients fibroblasts and *pdss2* mutant mice, we now investigated the H₂S oxidation pathway after 1) CoQ supplementation in patients fibroblasts, 2) pharmacological inhibition of CoQ biosynthesis in wild-type fibroblasts, 3) knock-down in Hela cells of ADCK3, a CoQ biosynthesis regulatory protein, 4) knock-down in Hela cells of SQR, 5) NaSH supplementation in wild-type fibroblasts.

Major Points

1. Increased protein sulfhydration, owing to failure of H₂S oxidation, is presumed by the authors to be a major basis for pathology associated with CoQ-depleting gene mutations. Experiments should be performed to determine whether CoQ levels are indeed inversely correlated with the extent of protein sulfhydration. Thus, it is recommended that the authors quantify the extent of protein sulfhydration in genetically CoQ deficient patient cell lines (using any of several reported proteomic methods) and determine the extent to which CoQ levels are negatively correlated with total protein sulfhydration and/or sulfhydration of specific proteins.

We postulate that defects of H₂S oxidation cause H₂S binding to *heme* moieties in proteins thereby inhibiting their activity. Consistent with our hypothesis, CoQ deficient mice showed abnormal acylcarnitine profile, indicating that the enzymatic activity of short-chain acyl CoA dehydrogenase (SCAD) is inhibited.

However, we agree with the reviewer that alterations of protein sulfhydration may be another pathomechanism associated with CoQ deficiency. We therefore quantified total protein S-sulfhydration in two cell lines with different degrees of CoQ deficiency, P1 (~50% residual CoQ and normal SQR) and P4 (~15% residual CoQ and ~25% residual SQR). We noted that sulfhydration was increased in both cell lines compared with controls. We have now added a new figure (Fig 7) in the manuscript reporting data on protein sulfhydration.

This result is consistent with results of SQR-driven respiration studies, that showed that SQR activity is also reduced in cell lines with normal SQR protein levels, although there is not strict correlation between SQR-driven respiration defect and the magnitude of protein sulfhydration. This may be due to 1) experimental conditions, since SQR-driven respiration was measured adding excess of NaHS as substrate, while protein sulfhydration was measured under native conditions, or 2) presence of other compensatory mechanisms regulating H₂S levels, for example in the upstream H₂S biosynthesis pathway, depending on CoQ levels or genetic background of the fibroblasts.

2. The authors report that SQR protein levels are reduced in proportion to CoQ10 levels in patient fibroblasts. Some consideration of the mechanistic basis for this phenomenon is needed. Does SQR loss arise from decreased SQR gene transcription, SQR protein translation, or accelerated SQR protein turnover? Is SQR loss driven by accumulation of sulfide in cells, i.e., can the effect be mimicked by chronic sulfide exposure of cells possessing control levels of CoQ10?

Our hypothesis is not that SQR loss is driven by accumulation of sulfides in cells. We postulate that CoQ deficiency leads to reduced SQR activity and protein instability, which in turn causes accumulation of sulfide. However, CoQ biosynthesis inhibition affects *SQR* gene expression. Our hypothesis is based on the following results:

- 1) SQR protein levels are reduced in proportion to CoQ levels in patient fibroblasts (Fig. 4), while SQR mRNA was significantly reduced in P1, P3 and P4 (Fig. EV1).
- 2) Inhibition of CoQ biosynthesis in wild-type fibroblasts by a pharmacological approach, using 4-NB, causes the same level of CoQ deficiency of P2. In both cases SQR protein levels are normal, but mRNA levels are increased, suggesting a compensatory mechanism (Fig. EV3A, B).
- 3) CoQ supplementation in patient fibroblasts increased SQR protein levels (Fig. EV2A and Fig. 5A, B), indicating an effect of CoQ on protein stabilization.
- 4) In Hela cells, knock-down of ADCK3, causes CoQ deficiency (~50% residual), and consistently, significantly reduces SQR mRNA and protein levels (Fig. EV3C and Fig.6).
- 5) Exposure of wild-type fibroblasts to 0.5 mM NaHS for 24h (previously shown to be enough to have toxic effects, Di Meo I, 2011) does not cause reduction of SQR levels (Appendix Fig.S3).

3. The authors demonstrate that exogenous CoQ10 can restore the ability of CoQ-deficient mutant cells to oxidize sulfide (Fig 3B). Surprisingly,

this finding is made after a one-week exposure to CoQ10. While it is inferred that the observed restoration of sulfide oxidation is due to CoQ10 repletion, another possibility is that SQR levels are restored by this long duration of CoQ10 treatment. It is recommended that SQR levels are quantified in CoQ10-treated cells to ascertain whether SQR abundance is restored by CoQ10 treatment.

We quantified SQR protein levels and mRNA in P3, P4, and control fibroblasts after CoQ supplementation and we observed that SQR protein levels were significantly increased in P3 and partially in P4 (Fig. 5 and Fig. EV2), indicating that restoration of sulfide oxidation is not entirely due to increased SQR.

Notably, it's surprising that CoQ10 repletion would require one week of incubation. Is this long incubation period necessary to replete cellular levels of CoQ10?

We supplemented fibroblasts with CoQ₁₀ for one week because the H₂S oxidation pathway enzymes are localized to mitochondria and CoQ pharmacokinetic to reach the mitochondria is delayed by its poor bioavailability, thus a long period of incubation is required for effective results. We previously showed that CoQ₁₀ cellular repletion happens after 24h of supplementation with 5 μ M CoQ₁₀; however, one week is necessary for CoQ₁₀ to reach the mitochondria and to improve mitochondrial bioenergetics, as measured by ATP production (Lopez LC, 2010). Thus, the delayed normalization of SQR levels after initiation of CoQ supplementation is consistent with the timeline of normalization of the mitochondrial respiratory chain.

4. Fig 4: While CoQ and SQR levels decrease to the same degree in both P3 and P4 cell lines, TST and ETHE1 protein levels are only significantly increased in the P4 cell line. Can the authors provide an explanation for this apparent inconsistency? Was P5 purposefully omitted in this figure?

P5 was not purposefully omitted. It was used only for the oxygen consumption experiment (Fig. 1, Appendix Fig.S1 and Appendix Fig.S2). It was not available for the other experiments. Our data suggest that CoQ regulates SQR levels, which triggers a response of the downstream pathway. In patients fibroblasts there is a trend toward up-regulation of the pathway. We can not account for the differences between cell lines, which might be due to genetic background. We excluded differences in ROS, or the presence of CoQ biosynthesis intermediates, as suggested by Reviewer 2.

5. Fig 7A: In both the Results and Discussion sections, the authors state that SQR was almost undetectable in brains of WT mice and the level highly increased in mutant animals. This is not apparent in the presented data. Indeed, SQR protein bands seem to be present with similar intensities. Also a p-value has not been indicated on the graph - is the observed difference statistically significant?

We measured the intensity of the bands of 5 WT and 5 MUT extracts repeated in 3 independent experiments. The intensity of the SQR band in mutant animals was 154% \pm 72 SD compared to controls. We agree that the difference was not statistically significant. To clarify this point, we changed the text from “exhibited a considerable increase” to “exhibited a trend toward increase”.

What is the basis for normalization of western blot findings to vinculin?

We used vinculin because it is a housekeeping protein whose size does not overlap with any of the proteins we wanted to test. Western blot of mice tissue extracts were also normalized to TOM20, a mitochondrial outer membrane protein, and the results were the same of the normalization to vinculin, therefore were not included.

6. The discrepancy between findings made with human fibroblasts and the mouse model of CoQ10 deficiency is not been adequately addressed in the Discussion section. Is it organism or tissue related? Please comment.

We added this comment to the Discussion: “We hypothesize that the discrepancy between findings in human fibroblasts and mouse kidney is mostly tissue-related. Tissue-specificity is typical of human and murine mitochondrial disorders, and fibroblasts are not clinically affected. We previously observed that Pdss2 mutant fibroblasts do not show the detrimental effects of CoQ deficiency observed in Pdss2 mutant mice kidney. It is possible that the molecular and biochemical abnormalities observed in patients fibroblasts reflect more severe abnormalities or a selective vulnerability to the effects of CoQ deficiency of the affected organs, for example kidneys, which is often affected in CoQ deficiency, independently of the molecular defect.”

Minor Points

Table 1: The CoQ level for P5 has been omitted from the table, presumably by accident. Please add this value.

We added the value, which was omitted by accident.

Fig 1. Labels for sulfide oxidation enzymes are provided as a black text on a dark background - barely legible. Please modify to improve legibility.

We changed the background to improve legibility.

Fig 5: On line 201 the text refers to Fig 5C, D. This should be corrected as 5A, B.

We apologize for the mistake. Figures were re-numbered in the revised manuscript.

References to supplementary figures between Lines 260-266 need to be corrected. (Figure S3 apparently refers to S4, S2 refers to S3 and S1 refers to S2).

We apologize for the mistake. Figures were re-numbered in the revised manuscript.

2nd Editorial Decision

19 October 2016

Please find enclosed the final reports on your manuscript. We are pleased to inform you that your manuscript is accepted for publication and is now being sent to our publisher to be included in the next available issue of EMBO Molecular Medicine.

***** Reviewer's comments *****

Referee #2 (Remarks):

The Authors have performed an excellent revision of the original manuscript by performing new experiments and answering in detail in a satisfactory way to all queries by this reviewer. The manuscript represents an important and novel contribution to the field. I recommend acceptance.

Referee #3 (Remarks):

The authors have satisfactorily addressed all prior reviewer's concerns with new findings and revised text. The manuscript now makes a compelling case for CoQ levels as a physiological determinant of the sulfide oxidation pathway activity and enzyme expression.

Corresponding Author Name: Catarina M Quinzii

Journal Submitted to: Embo Mol Med

Manuscript Number: EMM-2016-06356